# Spatial transcriptomics reveals regionally altered gene expression that drives retinal degeneration
Ulrike Schumann[1,2,3,8] ✉, Lixinyu Liu[1,2,4,5,8], Riemke Aggio-Bruce[1,6], Adrian V. Cioanca[1,6], Artur Shariev[1], Michele C. Madigan [3,7], Krisztina Valter[1,6], Jiayu Wen [1,2,4,5] ✉ & Riccardo Natoli [1,2,6] ✉

Photoreceptor cell death is a hallmark of age-related macular degeneration. Environmental, lifestyle and genetic risk factors are known contributors to disease progression, whilst at the molecular level, oxidative stress and inflammation are central pathogenetic drivers. However, the spatial and cellular origins of these molecular mechanisms remain unclear. We used spatial transcriptomics to investigate the spatio-temporal gene expression changes in the adult mouse retina in response to photo-oxidative stress. We identify regionally distinct transcriptomes, with higher expression of immunity related genes in the superior retina. Exposure to stress induced expression of genes involved in inflammatory processes, innate immune responses, and cytokine production in a highly localised manner. A distinct region ~800 µm superior from the optic nerve head seems a key driver of these molecular changes. Further, we show highly localised early molecular changes in the superior mouse retina during retinal stress and identify novel genes drivers. We provide evidence of angiogenic changes in response to photo-oxidative stress and suggest additional angiogenic signalling pathways within the retina including VEGF, pleiotrophin and midkine. These new insights into retinal angiogenesis pave the way to identify novel drivers of retinal neovascularisation with an opportunity for therapeutic development.

The retina is a highly laminated structure, consisting of alternating layers of neuronal cell bodies and processes arranged to capture light and relay this information to the visual cortex[1]. The main type of neurons found in the retina are photoreceptors, which are responsible for converting light signals into chemical signals, leading to electrical impulses sent to the brain. This process, referred to as phototransduction, is one of the most energy-intensive systems in mammals, putting the photoreceptors and the retina in a constant state of oxidative pressure, leading to oxidative stress, inflammation and consequently retinal degeneration[2–5]. While there has been significant work to explore the molecular factors underpinning retinal degenerations, to date no studies have mapped the full transcriptome in both a spatial and temporal manner.

The advent of quantitative transcriptome-wide spatial gene expression technologies in the 2010's, recognised by Nature as method of the year in 2020[6], has dramatically transformed our understanding of tissue architecture[7–10] and cellular interactions[11–13], especially in the tumor microenvironment[14–18], leading to new discoveries that promise the development of novel and targeted therapeutic approaches. As retinal degenerations often have highly localised pathological presentations, this novel technology allows the simultaneous exploration of gene expression changes at the site of damage and the surrounding penumbra, facilitating in depth investigations into where molecular dysregulations first occur. To our knowledge no study has mapped the spatio-temporal molecular changes of the healthy retina and in response to degeneration.

Here, we utilised the Visium Spatial Gene Expression technology (10X Genomics) to spatially resolve the gene expression changes that occur during a time-course of retinal degeneration using a mouse model of photo-oxidative damage[2]. We were able to identify transcriptional clusters

[1]The John Curtin School of Medical Research, The Australian National University, Canberra, Australia. [2]The Shine-Dalgarno Centre for RNA Innovation, The John Curtin School of Medical Research, The Australian National University, Canberra, Australia. [3]The Save Sight Institute, The University of Sydney, Sydney, Australia. [4]The Centre for Computational Biomedical Sciences, The John Curtin School of Medical Research, The Australian National University, Canberra, Australia. [5]ARC Centre of Excellence for the Mathematical Analysis of Cellular Systems (MACSYS), The Australian National University, Canberra, Australia. [6]The School of Medicine and Psychology, The Australian National University, Canberra, Australia. [7]The School of Optometry and Vision Science, The University of New South Wales, Sydney, Australia. [8]These authors contributed equally: Ulrike Schumann, Lixinyu Liu. ✉e-mail: Ulrike.Schumann@anu.edu.au; jiayu.wen@anu.edu.au; riccardo.natoli@anu.edu.au

representative of the retinal layers and we reveal distinct transcriptional profiles between the superior and inferior retina in dim-reared controls, including a naturally heightened immune response in the superior mouse retina. Further, we find progressive dysregulation in the superior transcriptome and identify a highly localised region, around 800μm superior of the optic nerve head, from which gene expression changes originate. By focussing our analysis specifically on this region, we uncover early upregulation of genes with known roles in pathological neovascularisation, suggesting that angiogenic pathways are altered in response to photo-oxidative stress. Cell communication analysis suggested that pro-inflammatory cytokine and angiogenesis signalling pathways (VEGF, midkine (MDK) and pleiotrophin (PTN)) play key roles in communication across retinal layers. Finally, incorporating single-cell sequencing data highlighted specific cell types that might facilitate this communication. Overall, by focusing on early stages of retinal degeneration, we identified several genes and pathways that are dysregulated prior to significant retinal pathology, identifying novel targets for future studies and potential therapeutic development.

## Results

We performed Spatial Gene Expression (Visium, 10X Genomics) to gain spatially resolved insights into the retinal transcriptional changes in a mouse model of photo-oxidative damage (PD)[2], which causes central superior photoreceptor degeneration. Histological analysis shows immune cell infiltration, progressive photoreceptor cell death (Supplementary Fig. S1A), and concomitant thinning of the photoreceptor layer specifically in the superior retina across a PD time course of 1 (1PD), 3 (3PD) and 5 days (5PD) compared to dim-reared (DR) controls (Supplementary Fig. S1B), as previously reported[2,19–22]. Tissue permeabilisation optimisation and spatial gene expression was performed following manufacturer's instructions (Supplementary Fig. S1C-D), and sequencing reads mapped to expression spot coordinates using SpaceRanger (10X Genomics). We assessed data quality by analysing RNA diffusion rates (SpotClean[23]; Supplementary Fig. S2A-B) or by a filtering approach (<500 features; >30% mitochondrial reads) to exclude spots with low expression levels. We retained 4077 spots across all 8 samples (2 each for DR, 1PD, 3PD and 5D), which were pooled to minimise batch effects. Reproducibility between the biological replicate samples is exceptional ($r \geq 0.98$; Supplementary Fig. S2C), showing minimal significant differential expression (Supplementary Data S1-Table S1, Supplementary Fig. S2D).

### Spatial gene expression profiles align with the functional and morphological organisation of the eye

Initial spot clustering using Seurat[24] yielded 20 distinct expression clusters (Supplementary Fig. S3A). Differential gene expression analysis across all clusters (Supplementary Fig. S3B) was then combined with visualisation of cluster position within the eye tissue, allowing us to assign ocular morphological features to all clusters (Supplementary Fig. S3A). However, due to the relatively large spot size (~55 μm) only ~31% of the spots covered the retinal tissue, limiting the spatial resolution (Supplementary Fig. S3C). Therefore, cell type-specific gene expression (Supplementary Data S1-Table S2) was used to further refine this morphological annotation (Supplementary Fig. S4). This shows that most spot clusters contained transcriptional contributions from multiple cell types, a limitation due to both the resolution and lack of including morphology into cluster identification.

Thus, tissue morphology was integrated with gene expression profiles during clustering using the stLearn and Harmony packages[25,26], which yielded 15 expression clusters (Fig. 1A, Supplementary Fig. S5A). Gene expression was distinct between clusters (Fig. 1B) and all 15 clusters were represented across all tissue sections (Fig. 1C). As above, the differential gene expression across all clusters (Supplementary Fig. S5B) was combined with cell type-specific gene expression (Supplementary Fig. S6) to assign ocular morphological features to each cluster (Supplementary Fig. S5A). This approach identified distinct clusters that clearly overlap with the four layers of the posterior eye; ganglion cells (C1), inner nuclear

layer (C4), photoreceptors (C2), and retinal pigment epithelium (RPE)/choroid (C8), which we termed L1-L4, respectively (Fig. 1Di). To improve our spatial understanding of the data, we plotted cell type-specific gene expression[27] firstly within each retinal/RPE/choroid spot as percentage of total gene expression (Fig. 1Dii, Supplementary Fig. S7) and, secondly as percentage of all spots in each of the four layers at each time point (Fig. 1Diii). This spatial transcriptome not only mirrors the distinct layered structure of the retina but also illustrates the progressive loss of photoreceptors in the superior retina with PD (Fig. 1Dii), reflecting the underlying pathology of this model. Gene ontology (GO) term enrichment of genes differentially expressed between the four layers, reflects the known functionality of the predominant cell type in each layer (Fig. 1E; Supplementary Data S1-Table S3). The neuronal layers L1 and/or L2 are associated with GO terms related to neuronal function (e.g. GO:0042391—'regulation of membrane potential', GO:0043269—'regulation of ion transport', GO:0050808—'synapse organisation'; Supplementary Data S1-Table S3A&C), whereas the photoreceptor layer L3 is associated with photoreceptor development and function (e.g. GO:0007601—'visual perception', GO:0009584—'detection of visible light', GO:0042461—'photoreceptor cell development'; Supplementary Data S1-Table S3E). The RPE/choroid layer L4 shows associations with angiogenesis (e.g. GO:0048514—'blood vessel morphogenesis', GO:0001944—'vasculature development') and tissue structure (e.g. GO:0030198—'extracellular matrix organisation', GO:0031589—'cell-substrate adhesion'; Supplementary Data S1-Table S3G), reflecting their support function. Finally, we analysed differential gene expression between the entire superior and inferior retina across all samples and identified 145 differentially expressed genes (Fig. 1F, Supplementary Data S2-Table S4) including *Vax2*, *Nr2f2*, *Efnb2*, *Igf2b2*, *Opn1sw*, *Opn1mw* and *Abi3bp*, which have previously been shown to be significantly differentially expressed between the superior and inferior retina[28]. This shows that our data represents the morphological organisation and functional roles of distinct retinal structures.

### The transcriptional profile of the superior retina is primed to combat inflammation

Structural differences in RNA expression could be considered as drivers of degeneration. Therefore, differential gene expression between the superior and inferior retina at each time point (DR, 1PD, 3PD and 5PD: superior vs inferior) was analysed to further refine expression differences. Only seven genes (*Efnb2*, *Nr2f2*, *Vax2*, *Aldh1a1*, *Il33*, *Clec3b* and *Vax2os*; Fig. 2A, Supplementary Data S2-Table S4) are significantly differentially expressed between superior and inferior retinal regions at all four time points (Fig. 2B), with an additional 17 genes differentially regulated in three out of the four time points (Fig. 2A). Interestingly, the majority of these genes (19/24), including *Enpp2*, *Lpar1*, *C1s1*, *C3* and *Il33* amongst others, are consistently upregulated in the superior retina across the entire PD time course (Fig. 2B, Supplementary Data S2-Table S4). To gain a clearer insight into the inherent functional requirements of the superior retina, we performed GO term analysis of all genes significantly upregulated in the superior retina at DR with redundancy reduction of significantly enriched (FDR ≤ 0.05) GO terms using REVIGO[29]. Upregulated genes in the superior retina are associated with 'antigen processing and presentation' terms (GO:0019886) and 'immunoglobulin mediated immune response' terms (GO:0016064; Fig. 2C, Supplementary Data S2-Table S5A). This includes several H-2 class II histocompatibility antigen genes (*H2-Eb1*, *H2-Ab1*, *H2-Aa*) and the class II histocompatibility antigen gamma gene *Cd74*. Additionally, upregulated genes are also associated with 'extracellular matrix/structure organisation' terms (GO:0030198, GO:0043062), including the extracellular matrix protein dermatopontin (*Dpt*) and collagen alpha chains (*Col1a1*, *Col3a1*).

To determine the molecular changes that occur in the superior retina in response to increasing oxidative stress and inflammation compared to the inferior retina, we examined each PD time point individually (1PD, 3PD and 5PD: superior vs inferior). At 1PD, genes upregulated in the superior show a significant association with 'lens development of camera-type eye'

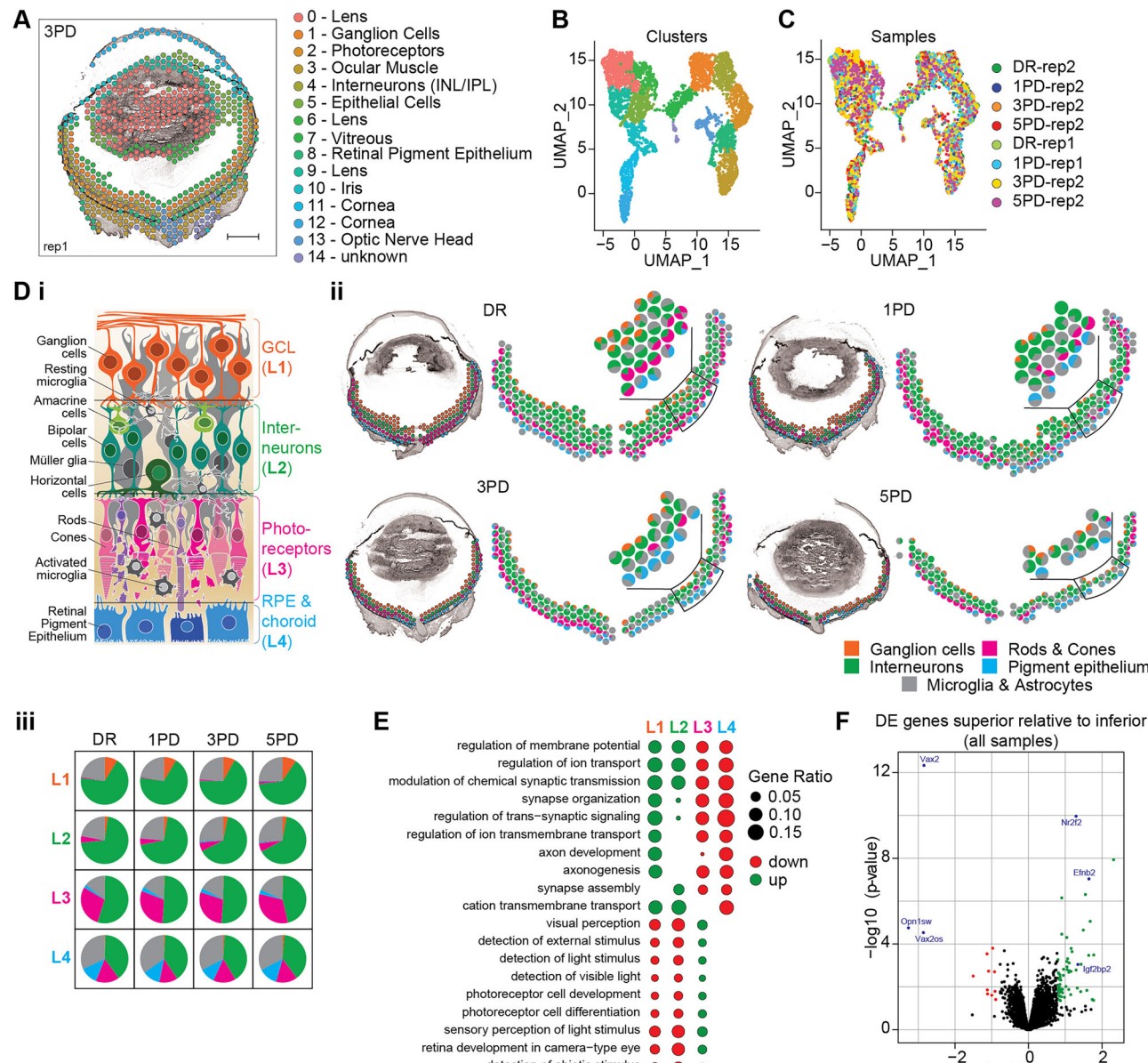

**Fig. 1 | Spatial gene expression profiles align with the functional and morphological organisation of the eye. A** Unsupervised clustering incorporating morphological information from the H&E images using stLearn[25]. Sample 3PD replicate 1 is shown as representative, all other samples are shown in Supplementary Fig. S5. Spots are coloured according to their cluster. Scale bar = 500 μm. **B, C** Reduced dimension presentation (UMAP) of the identified clusters coloured (B) according to clusters or (C) according to sample of origin. **D** stLearn produced exceptional resolution of the distinct architecture of the posterior mouse eye with four clusters representing the four main retinal/choroid layers. (*i*) Schematic illustrating a cross section of the retina including ganglion cell layer (L1, GCL, orange), interneurons (L2, green), photoreceptors (L3, pink), and retinal pigment epithelium (RPE) and choroid (L4, RPE/choroid, blue). (*ii*) Spots within the four retinal/choroid clusters coloured according to the layering L1-4 (left). Scatter pie plots showing percentage

expression of retinal cell type-specific marker genes[27] for each spot within the four layers (right). Spot cluster overlap with retinal/choroid layers and scatter pie plots for all eight samples are shown in Supplementary Fig. S8 and S7, respectively. (*iii*) Pie charts showing percentage of cell type-specific marker gene expression in each of the four retinal/choroid layers at each time point. (**E**) GO term enrichment of significantly differentially expressed genes in each of the four layers across all eight samples. GO terms enriched in at least three layers are shown, all enriched GO terms are listed in Supplementary Data S1-Table S3. **F** Volcano plot showing differential gene expression in the superior retina relative to the inferior retina across all eight samples. Significantly differentially expressed genes ($p \leq 0.05$, FC $\geq 1.5$; Supplementary Data S2-Table S4) are coloured green (upregulated) and red (downregulated) with genes previously identified as differentially expressed[28] labelled in blue.

(GO:0002088), 'visual perception' (GO:0007601) and 'sensory system/organ development' (GO:0048880, GO:0007423) GO terms compared to the inferior retina (Fig. 2C, Supplementary Data S2-Table S5B). Interestingly, genes contributing to these pathways are mainly part of the crystallin gene families (Supplementary Data S2-Table S5B). At 3PD, genes upregulated in the superior retina are associated with pathways that indicate degeneration (e.g. GO:0031295–'T cell co-stimulation', GO:0048514–'blood vessel morphogenesis', GO:0030198–'extracellular matrix organisation', GO:0043062–

'extracellular structure organisation'; Fig. 2C, Supplementary Data S2-Table S5C). At 5PD genes upregulated in the superior are associated with multiple pathways relating to inflammation and immune response (e.g. GO:0006954–'inflammatory response', GO:1903555'–regulation of tumour necrosis factor superfamily cytokine production', GO:0001816–'cytokine production', GO:0050776–'regulation of immune response', GO:0000 2526–'acute inflammatory response'; Fig. 2C, Supplementary Data S2-Table S5D). Together this pattern indicates that relative to the inferior

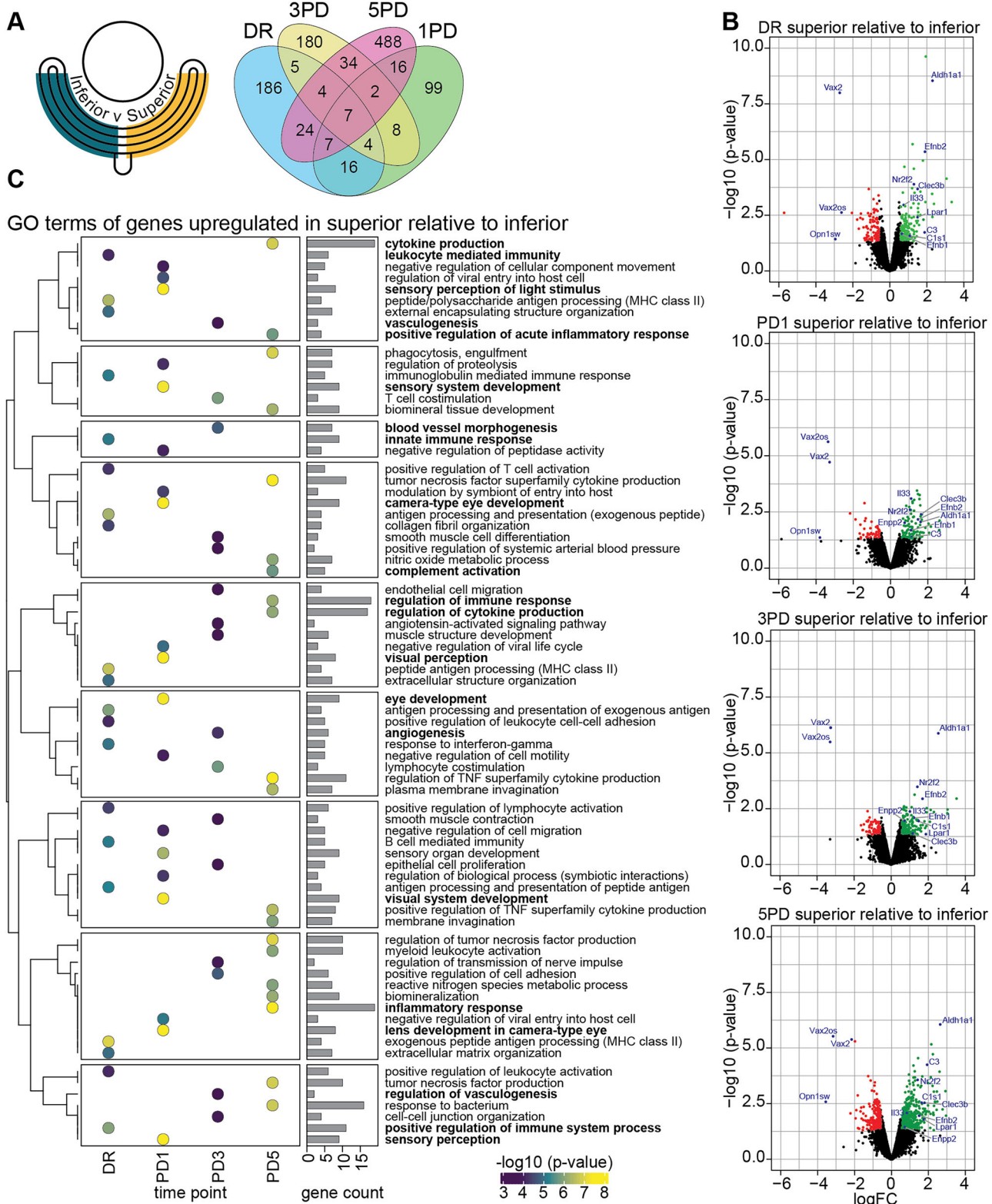

**Fig. 2 | The transcriptional profile of the superior retina is primed to combat inflammation. A** Schematic showing the gene expression analysis paradigm: DR, 1PD, 3PD and 5PD: superior vs inferior (left). Venn diagram showing the overlap of significantly differentially expressed genes in the superior at each time point (right; $p \le 0.05$, FC $\ge 1.5$; Supplementary Data S2-Table S4). **B** Volcano plots showing differentially expressed genes in the superior at each time point with significant genes coloured green (upregulated) and red (downregulated). Genes previously identified as differentially expressed[28] and genes with known roles in the stress response are highlighted in blue. **C** GO term enrichment analysis (Supplementary Data S2-Table S5) of genes significantly upregulated in the superior at each time point. The top significantly enriched terms are shown (FDR $\le 0.05$).

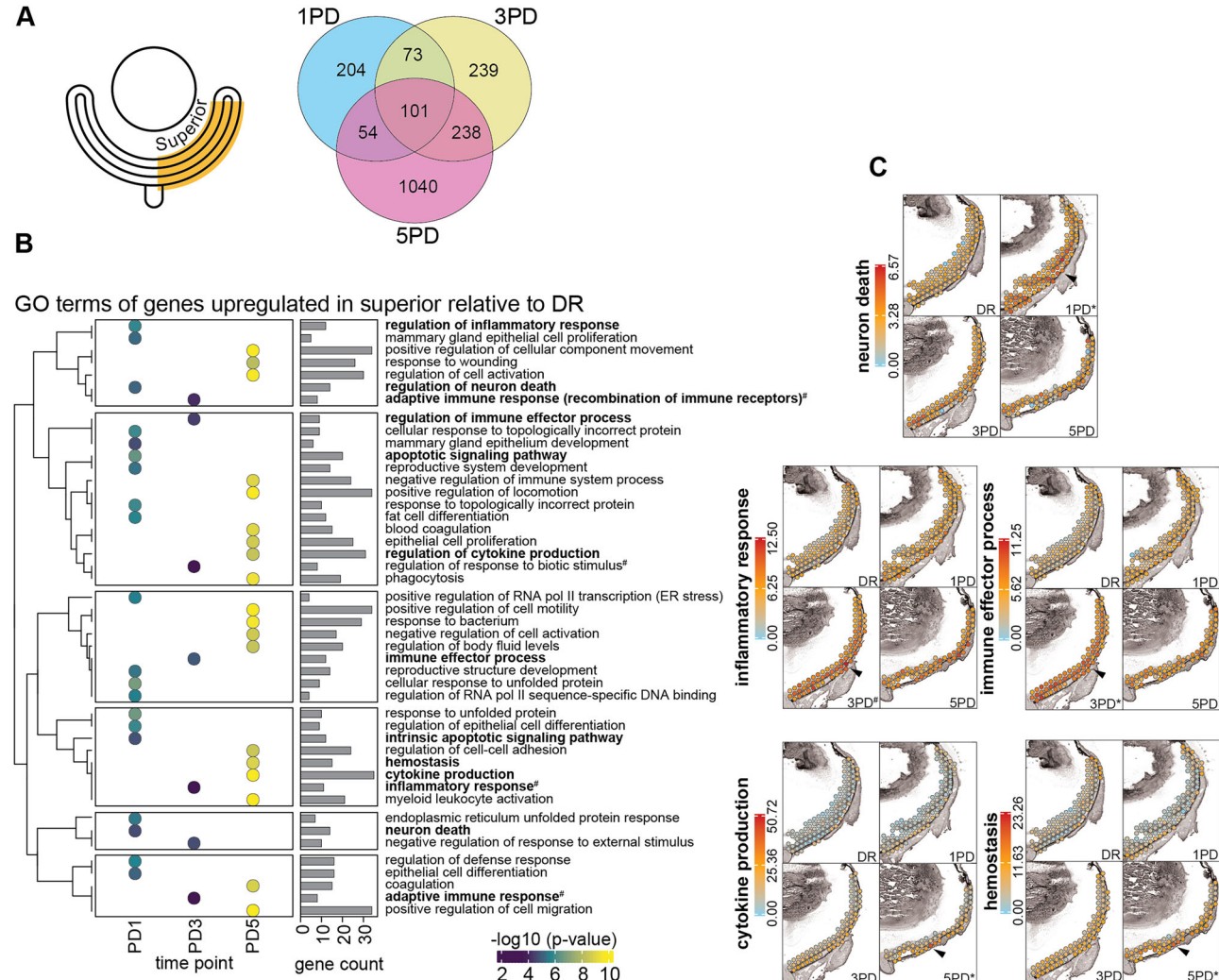

**Fig. 3 | Progressive gene expression changes in the superior retina establishes the retinal immune response pattern. A** Schematic showing the gene expression analysis paradigm: superior only: 1PD / 3PD / 5PD vs DR (left). Venn diagram showing overlap of differentially expressed genes (right; $p \leq 0.05$, FC ≥ 1.5; Supplementary Data S3-Table S6A). Volcano plots of differential expressed genes are shown in Supplementary Fig. S9A. **B** GO term enrichment analysis (Supplementary Data S3-Table S7) of significantly upregulated genes. The top enriched terms are shown (FDR ≤ 0.05). #indicates terms not significantly enriched (FDR ≥ 0.05, $p \leq$ 0.05). **C** Per spot cumulative expression levels (indicated by the heatmaps) of all significantly upregulated genes contained in indicated GO term for spots overlapping the superior retina. Asterisk indicates the time point where GO term is significantly enriched (FDR ≤ 0.05); #indicates terms not significantly enriched (FDR ≥ 0.05, $p \leq$ 0.05). One representative replicate for each time point is shown. Arrowheads indicate region with strongest upregulation.

retina, the superior retina undergoes extensive molecular changes in response to chronic inflammation at 5PD. Interestingly, genes upregulated at 5PD are also associated with phagocytic terms (GO:0099024, GO:0010324, GO:0006909) and extracellular matrix/structure organisation terms (GO:0030198, GO:0043062), which might be indicative of either increased photoreceptor outer segment recycling[30,31] or retinal neuronal remodelling following acute light damage[22]. The progressive loss of photoreceptors in the superior retina at 5PD is reflected by the association of downregulated genes with 'visual perception' (GO:0007601), 'detection of light stimulus' (GO:0009583) and 'retina development in camera-type eye' (GO:0060041) terms (Supplementary Data S2-Table S5E). Overall, this data shows that prolonged PD results in an extensive transcriptional shift in the superior retina relative to the inferior retina.

### Progressive gene expression changes in the superior retina establishes the retinal immune response pattern

While transcriptional changes in the superior retina, relative to the inferior retina, were associated with chronic inflammation and established between 3PD and 5PD, it remains unclear precisely when and where in the superior retina these changes are initiated. To answer this, we analysed the gene expression changes that occur exclusively in the superior retina across the PD time course (superior only: 1PD / 3PD / 5PD vs DR). We identify 432, 651 and 1433 significantly differentially expressed genes at 1PD, 3PD and 5PD, respectively (Fig. 3A, Supplementary Fig. S9A, Supplementary Data S3-Table S6A), with 101 genes differentially expressed at all three time points when compared to DR controls. GO analysis of all significantly upregulated genes with REVIGO redundancy reduction was performed to understand the related pathways.

Typically, the progressive nature of the PD model means that few morphological changes can be detected at 1PD, although retinal metabolic stress can be detected very early[22]. It is therefore surprising that genes upregulated in the superior retina at 1PD are enriched in degenerative terms (e.g. GO:0097190–'apoptotic signalling pathway', GO:1901214–'regulation of neuron death'; Fig. 3B, Supplementary Data S3-Table S7A). Genes upregulated at 1PD are also associated with various stress response process terms (e.g. GO:0006989–'response to unfolded protein', GO:0035966 –'response to topological incorrect protein', GO:0031347–'regulation of defence response'). Interestingly, at 3PD upregulated genes in the superior retina are associated

with just three GO terms comprising GO:0002252–'immune effector process', GO:0002697–'regulation of immune effector process' and GO:0032102–'negative regulation of response to external stimuli', suggesting that gene activation is focussed on an immune response (Supplementary Data S3-Table S7B). However, at 5PD upregulated genes are highly significantly associated with >500 pathways (Supplementary Data S3-Table S7C). These include 'myeloid leukocyte activation' (reduced GO:0002274), 'inflammatory response' (GO:0006954), 'positive regulation of cell migration' (GO:0030335), and several cytokine-related terms (GO:0001816, GO:1903555, GO:0071706) among many others. Other significant terms of interest include phagocytotic terms (GO:0006909, GO:0050764), coagulation terms (GO:0007596, GO:0050817, GO:0050818), and 'blood vessel morphogenesis' (GO:0048514). Structural terms (GO:0061448—'connective tissue development', GO:0031589—'cell-substrate adhesion', GO:0050678—'epithelial cell proliferation regulation') were also significantly associated with upregulated genes.

We also identified a significant number of genes upregulated specifically in the inferior retina across the time course (inferior only: 1PD / 3PD / 5PD vs DR; Supplementary Data S3-Table S6B). Interestingly, upregulated genes are significantly associated with cytokine processes (e.g. GO:0001819—'positive regulation of cytokine production', GO:0001816—'cytokine production') across the entire time course (Supplementary Fig. S9B, Supplementary Data S3-Table S8). Additionally, genes upregulated in the inferior retina at 1PD are associated with GO:0032620–'interleukin-17 production' (Supplementary Data S3-Table S8A). At 3PD several immunity pathways are enriched (e.g. GO:0045087–'innate immune response', GO:0002888–'positive regulation of myeloid leukocyte mediated immunity', GO:0050729–'positive regulation of inflammatory response'; Supplementary Data S3-Table S8B). However, the largest changes are observed at 5PD with upregulated genes significantly associated with 'immune effector process' (GO:0002252), 'regulation of immune response' (GO:0050776), and 'inflammatory response' (GO:0006954) but also 'phagocytosis' (GO:0006909), 'positive regulation of response to external stimulus' (GO:0032103), 'leukocyte proliferation/differentiation' (GO:0070661, GO:0002521), and 'chemokine production' (GO:0032602) amongst others (Supplementary Data S3-Table S8C). This shows that there are extensive transcriptional changes at the early stages of PD, especially within the immune landscape, in the inferior retina although phenotypic changes are minimal. This rapid and extensive activation of the immune response suggest that the inferior retina might be primed to protect against oxidative stress and inflammation, especially since overhead visual information is processed in this region[32].

To explore whether there are specific spatial transcriptional patterns that underlie these molecular pathway changes, we plotted the cumulative expression level of all upregulated genes contained within significantly enriched GO terms. We analysed the following pathways: GO:0070997—'neuron death' (enriched at 1PD), GO:0006954—'inflammatory response' (at 3PD, *n.s.*), GO:0002252—'immune effector process' (enriched at 3PD), GO:0001816—'cytokine production' (enriched at 5PD), and GO:0007599—'hemostasis' (enriched at 5PD). Focusing on the superior retina, we found that expression of all GO term enriched, upregulated genes is concentrated in a small region, approximately 800μm from the optic nerve head (Fig. 3C; arrow heads). It has previously been suggested that the PD paradigm evokes greater insult to this central superior retinal region due to an increased density of photoreceptors compared to the periphery[31], especially of L-opsin-expressing cones in pigmented mice[33]. This difference in photoreceptor distribution results in an increased phagocytic load for the RPE and consequently higher susceptibility to oxidative stress and inflammation. Contrasting, the cumulative expression of genes upregulated in the inferior retina and enriched in the key GO terms GO:0001816—'cytokine production' (enriched at 1PD), GO:0045087—'innate immune response' (enriched at 3PD), and GO:0002252—'immune effector process' (enriched at 5PD) was not concentrated within a distinct region

but spread across the entire inferior retina (Supplementary Fig. S9C). Although similar GO terms are enriched in upregulated genes in both the inferior and superior retina, the specific genes contained within these pathways (Tables S10-11), as well as the time trajectory and magnitude of upregulation and enrichment, differs between both regions, suggesting distinct biological outcomes.

## The central superior retina shows distinct transcriptional changes and susceptibility to photo-oxidative impact

The focally localised nature of the transcriptional changes in the superior retina (~800 μm from optic nerve head) prompted us to ask specifically how the transcriptional profile in this location differs from the neighbouring regions. To address this question, we segmented the whole retina into six roughly equidistant regions (Fig. 4A, Supplementary Fig. S10A), with the central superior region of interest termed region five (R5). Differential gene expression analysis within each region across the time course (R1-R6: 1PD / 3PD / 5PD vs DR) shows that R5 has the most dramatic gene expression change of all six regions, with the highest number of dysregulated genes identified at 5PD (Fig. 4A, Supplementary Data S4-Table S9). Genes significantly upregulated in R5 at 1PD were similar to those observed for the whole superior retina at 1PD, suggesting that this area is the initial source of the transcriptional stress response to counter cell death in response to oxidative stress (Supplementary Data S3-Table S7, Supplementary Data S4-Table S10). Upregulated genes are significantly associated with neuronal apoptotic processes (GO:1901215—'negative regulation of neuron death', GO:0070997—'neuron death', GO:1901214—'regulation of neuron death'), transcription terms (GO:0043618—'regulation of transcription from RNA PolII promoter in response to stress', GO:0043620—'positive regulation of transcription from RNA PolII promoter in response to stress'), and oxidative stress processes (GO:1902882—'regulation of response to oxidative stress', GO:1900407—'regulation of cellular response to oxidative stress'; Fig. 4B, Supplementary Data S4-Table S10A). Interestingly, both neuronal death terms (GO:0070997, GO:1901214) associated with genes upregulated at 1PD in R5, are also associated with genes upregulated at 5PD when analysing the whole superior retina, although the number of genes contained within these differs (Supplementary Data S4-Table S10A, Supplementary Data S3-Table S8C). At 3PD, the transcriptional changes in R5 are clearly geared towards an immune response, with upregulated genes significantly enriched within immune-related GO terms (GO:0050777—'negative regulation of immune response', GO:0002252—'immune effector process'; Fig. 4B, Supplementary Data S4-Table S10B). At 3PD both these immune pathways were not significantly enriched when analysing the superior retina overall (FDR > 0.05, P < 0.05; Fig. 3B), but analysis of R5 in isolation revealed that a significant immune response occurs in a very localised manner. Additionally, significantly upregulated genes are already associated with 'regulation of angiogenesis' (GO:0045765) at 3PD in R5 (Fig. 4B), whereas this is only observed at 5PD when analysing the whole superior retina (Fig. 3B, Supplementary Data S4-Table S10B, Supplementary Data S3-Table S7C). At 5PD, similar to the whole superior retina, genes upregulated in R5 are highly significantly associated with GO terms related to cytokine production (GO:0001817—'regulation of cytokine production', GO:0001816—'cytokine production'), immune pathways (GO:0006954—'inflammatory response', GO:0050776—'regulation of immune response', GO:0032103—' positive regulation of response to external stimulus'), and 'vasculature development' (GO:0001944). Interestingly, genes upregulated in R5 at 5PD are also associated with cell/leukocyte migration (GO:0030335—'positive regulation of cell migration', GO:0050900—'leukocyte migration') and 'phagocytosis' (GO:0006909) terms (Supplementary Data S4-Table S10C).

The per spot cumulative expression of upregulated genes associated with GO terms of interest was plotted, showing that not only is gene upregulation strong in R5 overall but that some GO terms show even more localised upregulation within R5, especially at 5PD (Fig. 4C, arrow heads). For example, 'regulation of neuron death' (GO:1901214) is significantly associated with upregulated genes in R5 at 1PD and 5PD but not at 3PD. Similarly, 'cytokine production' (GO:0001816) is significantly

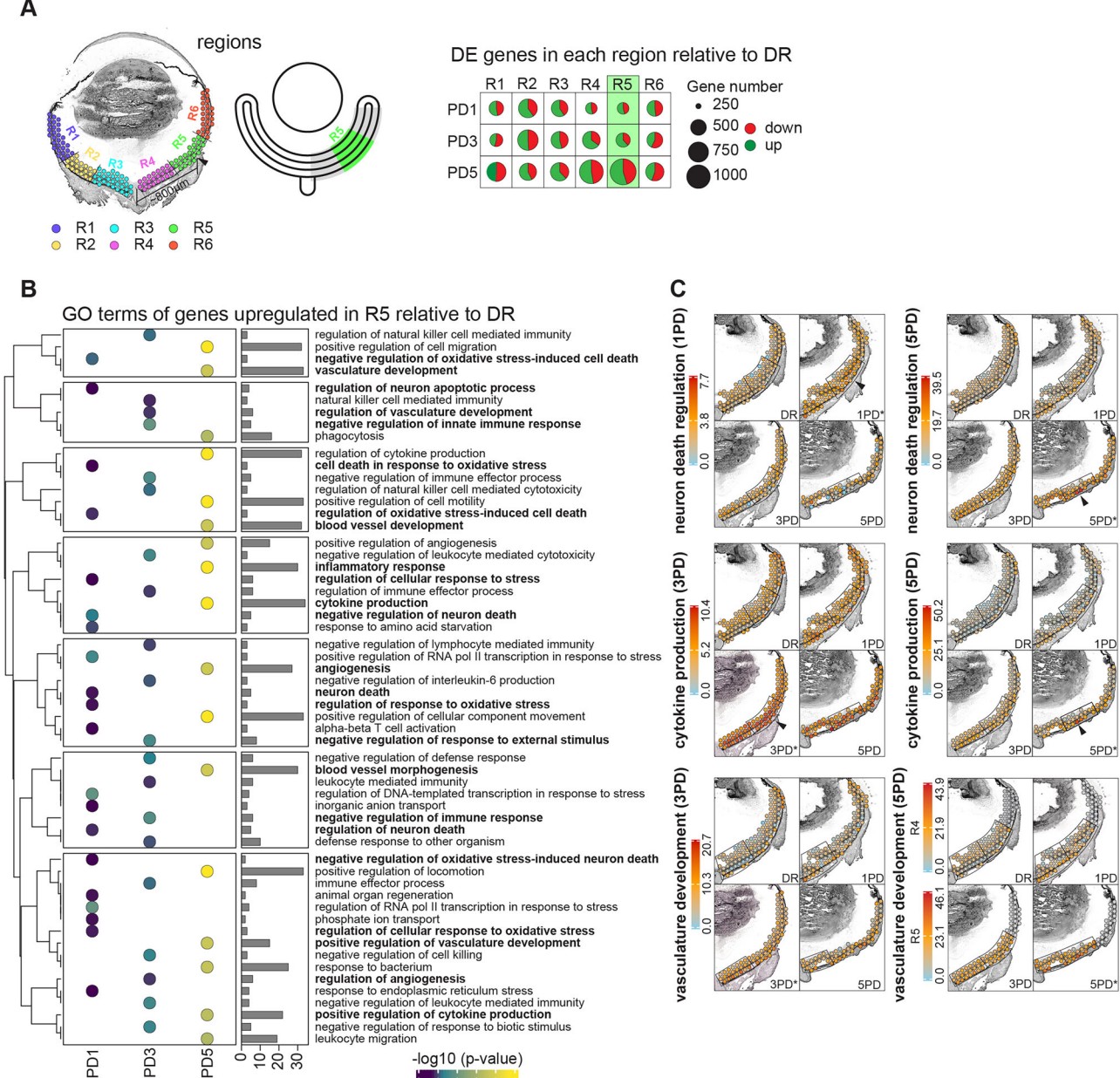

**Fig. 4 | The central superior retina shows distinct transcriptional changes to adapt to photo-oxidative impact. A** Representative sample illustrating the segmentation of retinal/choroid spots into six roughly equidistant regions (R1-R6; left). Arrowhead indicates the area with highest GO term enrichment approximately 800 µm superior from the optic nerve head. The segmentation for all samples is shown in Supplementary Fig. S10A. Schematic showing the gene expression analysis paradigm: R1-R6: 1PD / 3PD / 5PD vs DR (middle). Pie charts showing the significantly differentially expressed genes ($p \leq 0.05$, FC $\geq 1.5$; Supplementary Data S4-Table S9) in each region at each time point relative to the DR control. **B** GO term enrichment analysis (Supplementary Data S4-Table S10A-C) of significantly upregulated genes in R5. The top enriched terms are shown (FDR $\leq 0.05$). **C** Per spot cumulative expression levels (indicated by the heatmaps) of all significantly upregulated genes contained in indicated GO term for spots overlapping the superior retina. Asterisks indicates the time point where GO term is significantly enriched (FDR $\leq 0.05$). One representative replicate for each time point is shown. Arrowheads indicate region with strongest upregulation. Boxed regions indicate R5 of the superior retina.

enriched at 3PD and 5PD but not 1PD. In both cases the number and identity of the associated upregulated genes are distinct between the two time points (Supplementary Data S4-Table S10A-C). Further, we were interested to investigate the per spot expression pattern of the 'vasculature development' (GO:0001944) GO term following the discovery here that associated genes are strongly upregulated in our model. We discover an interesting pattern (Supplementary Data S4-Table S10, Supplementary Data S4-Table S11) with genes in this term initially downregulated in R5 at 1PD (Supplementary Fig. S10C), followed by an upregulation in R4 at 3PD and a further increase and expansion in expression to encompass R4 and R5 at 5PD (Fig. 4C). The latter is driven by the upregulation of a

much larger number of genes (Supplementary Data S4-Table S10, Supplementary Data S4-Table S11) which is again, highly localised within R5 (Fig. 4C, arrowhead).

Although we identified genes that are downregulated in R5 across the time course, only at 1PD were these significantly associated with any GO terms (Supplementary Data S4-Table S10D-F). As described above, downregulated genes in R5 across the time course showed significant associations with 'blood vessel development' (GO:0001568) as well as Rho signalling (GO:0035025—'positive regulation of Rho protein signal transduction', GO:0007266—'Rho protein signal transduction'; Supplementary Fig. S10B; Supplementary Data S4-Table S10D). Together, this analysis

demonstrates that R5 has a highly distinct gene expression profile when compared to other retinal regions in response to photo-oxidative damage.

## Gene expression changes originate in the inner retina and are mediated by key signalling networks

When analysing R5, we noticed that upregulated genes within the cytokine GO term changed expression intensity in a layer-dependent manner, with localised upregulation observed within the RPE/choroid at 5PD (Fig. 4C, middle row). To analyse this further and delineate the cell types contributing to this gene expression pattern, we analysed expression changes within each of the four retinal layers (superior L1-L4: 1PD / 3PD / 5PD vs DR) across the time course. As the spot numbers were insufficient to analyse layer-specific expression changes in R5 alone, we focussed on each layer within the superior retina as a proxy (Fig. 5A, Supplementary Data S5-Table S12). We performed GO term enrichment analysis of differentially expressed genes (Tables S17-18), focussing on key pathways we identified to be dysregulated in response to PD in this study. We found that initial dysregulation of genes occurs in L1 (ganglion cells) and slowly progresses across all remaining layers concurrent with progression of PD (Fig. 5A). Of note are inflammation-associated GO terms: 'immune effector process' (GO:0002252), 'response to cytokine' (GO:00034097), 'inflammatory response' (GO:0006954), and 'angiogenesis' (GO:0001525), which are associated with upregulated genes. Not surprisingly, visual perception 'detection of light stimulus' (GO:0009583) and 'sensory perception' (GO:0007600) terms were progressively downregulated across the time course with L2 (interneurons) and L3 (photoreceptors) most affected. Mixed patterns of up- and downregulation was observed, especially for genes associated with 'sensory organ development' (GO:0007423) but also for 'tissue homeostasis' (GO:0001894). Interestingly, a small number of genes associated with 'extracellular matrix organisation' (GO:0030198) are upregulated in L1 and L3 across the paradigm, but a much larger number is downregulated in L4 (RPE/choroid) at 3PD. These downregulated genes (Supplementary Data 5-Table S13) include *Ccn2*, which modulates many cellular aspects including cell adhesion and motility, and blood vessel formation and regeneration[34], and several collagen genes (type I, III, IV) involved in maintaining Bruch's membrane[35].

To understand whether this progressive gene expression pattern is due to changes in communication between retinal cell types in response to degeneration, we used CellChat[36] to model the probability of cell-to-cell communication across retinal layers. The top communication pathways with the largest number of ligand-receptor interactions across all layers and time points include VEGF, MDK (midkine), PTN (pleiotrophin) and PSAP (prosaposin), which all show distinct patterns of incoming (sink) and outgoing (source) signalling strength (Fig. 5B, Supplementary Fig. S11). Interestingly, the number of ligand-receptor interactions across all four layers for these four pathways was highest at 1PD, whereas at 3PD other signalling pathways additional to these four show large interaction numbers (Supplementary Fig. S11). The number of ligand-receptor interaction for VEGF, MDK, PTN and PSAP in DR control conditions range between five and 25. At 1PD, signalling within all four pathways is dramatically increased ranging between 35 and 50 interactions, whereas by 3PD and 5PD this is diminished to below ten interactions (Fig. 5B, Supplementary Fig. S11). While each layer serves as sink and source, the communication pattern for each layer is distinct across pathways (Fig. 5B, Supplementary Fig. S11). For example, L1 (ganglion cells) is a source of relatively high signalling strength of the VEGF and PTN pathways but L1 is not a source of MDK signalling and a relatively weak sink for all three pathways. L2 (interneurons) serves as both a source and sink for VEGF and PTN but as a sink only for MDK and PSAP signalling, demonstrating a central communication role for L2. L3 (photoreceptors) has the least relative signalling strength for all three pathways with equal signalling contribution as a source and sink. L4 (RPE/choroid) has a pattern opposite to L2 as it appears to be the primary source for MDK and PSAP signalling.

To better understand the communication between the four layers at each time point, we plotted the inferred signalling networks to visualise the strength of communication flow (Fig. 5C, Supplementary Fig. S12A). VEGF signalling originates from all layers and flows into L2 (interneurons) with L2 also serving as a source of VEGF signalling to L1. The signalling probability shows minimal change from DR conditions to 1PD, whereas a strong increase occurs at 3PD, especially from L2 to L1, with signalling diminishing at 5PD. PTN and MDK networks show different patterns with L1 and L2 both a source for PTN signalling to all other layers, whereas L4 is the primary source for MDK signalling to all other layers. Both networks show a stark increase in signalling probability at 1PD staying high at 3PD before diminishing at 5PD. The PSAP signalling network shows similarities to the VEGF network with L2 the major sink for signalling from all other layers but additional signalling from L4 to L1 (Supplementary Fig. S12A). There are no strong alterations in signalling strength across the time course for this network. To identify the main ligand-receptor pairs contributing to these networks, we plotted their communication probability for each possible direction of information flow across all four pathways. We found that most of the signalling between layers is mediated by just a few ligand-receptor pairings. Most pairings show no or very subtle changes in communication probability across the PD paradigm, so we focussed on those pairs with the most pronounced changes. We only identified a single receptor to interact with PSAP with minimal changes in communication probability across the time course (Supplementary Fig. S12B). Similarly, changes in communication probability of the VEGF ligand are subtle across the time course, but we observed a stronger increase in communication probability when the signal flow is directed towards L2, supporting the role of L2 as a sink (Fig. 5D). The increase of PTN communication probability from L1 and L2 at 1PD and 3PD is mediated specifically by the PTN-NCL pair, whereas the increase in MDK communication probability from L4 at 1PD and 3PD is mediated by the MDK-NCL pair. Interestingly, both PTN and MDK ligands also show interactions with the SDC receptors in L1/L2 and L4, respectively, but only PTN interacts with SDC3. Additionally, some of these pairings contribute to the signalling flow only at specific time points (Fig. 5D). For example, the PTN-SDC3 interaction is not detected at 3PD and the interaction of both MDK and PTN ligands with the NCL receptor is only present at 1PD and 3PD. Further, PSAP-GPR37 shows additional communication only at 1PD (Supplementary Fig. S12B), highlighting the complexity of retinal cytokine signalling. Overall, this data reveals changes in communication probability and signalling strength especially of the VEGF, MDK and PTN ligand-receptor pairs, with progressive dysregulation at 1PD and 3PD across all layers.

## Signalling network alterations are mediated by key retinal cell types

To understand whether the observed signalling pathways changes are mediated by expression changes, we plotted the differential gene expression of key ligands and receptors within each layer across the time course (Fig. 6A). Significant expression changes occur, but these are highly restricted to specific layers and/or time points. For example, the expression of *Vegfb* and *Ptn* is strongly downregulated in L1 at 3PD and L2 at 1PD, respectively. Only *Vegfr2* (*Kdr*) and *Sdc3* are differentially expressed in several layers at different time points, with *Vegfr2* (*Kdr*) upregulated at 5PD in all layers (except L2) and *Sdc3* dynamically expressed across the retina and time course (Fig. 6A).

To better understand these differences between layers, we asked which cell types might express these receptors and/or ligands and thus might be involved in the cross-retinal communication of these three signalling pathways. We utilised publicly available single-cell sequencing data[37] to extract receptor-ligand expression patterns (Fig. 6B, Supplementary Fig. S12C) and further plotted their (cumulative) per spot expression levels in the superior retina across the time course using our spatial data (Fig. 6C). *Vegfa* is strongly expressed in astrocytes and Müller glia with *Vegfb* showing low expression in these same cells. Additionally, spatial expression analysis showed preferential *Vegfa* expression in spots overlapping with the interneuron layer (L2) in the DR control, with expression diminished at 5PD. Both receptors (*Vegfr1* and *Vegfr2*) are

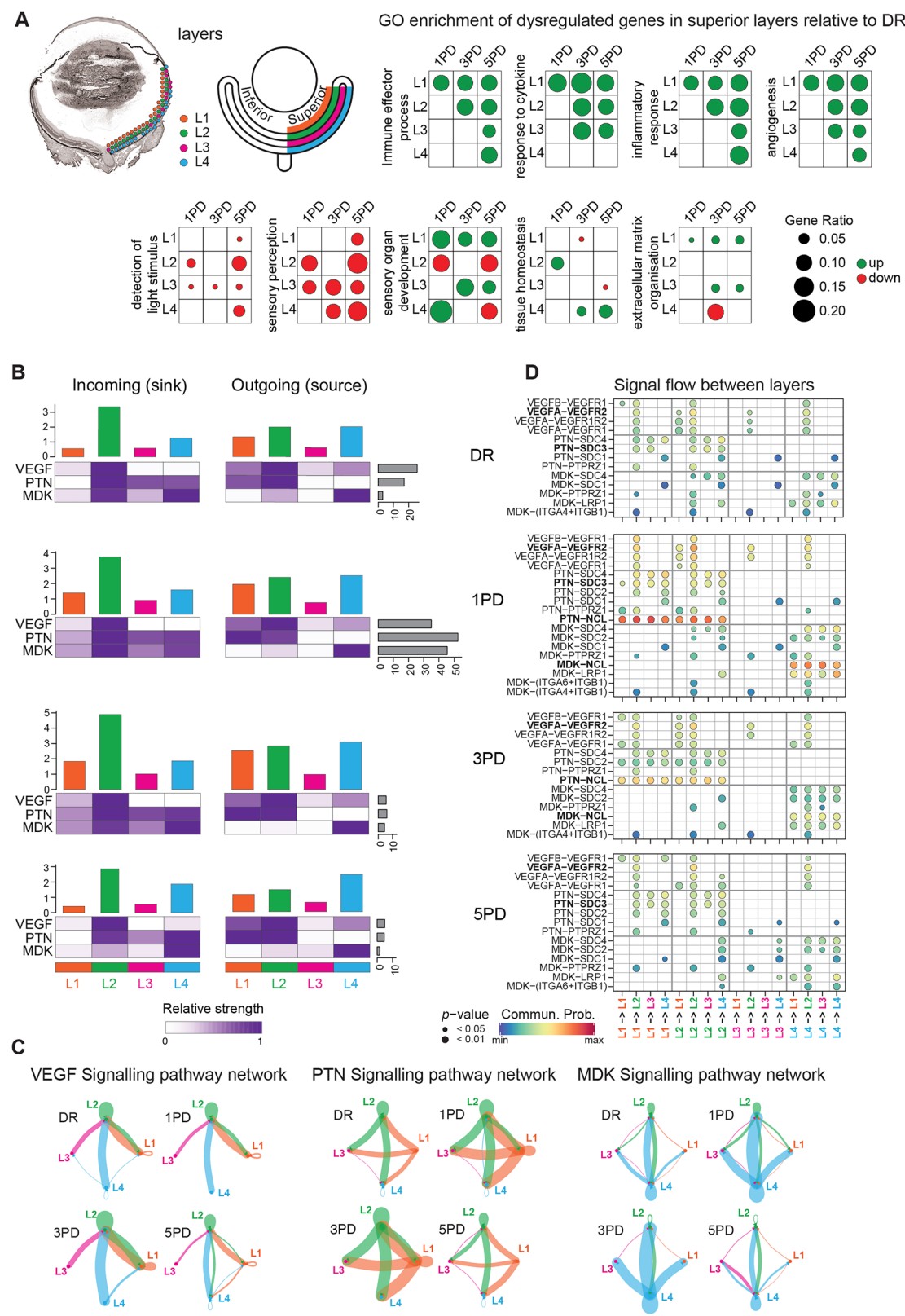

strongly expressed in Müller glia and vascular endothelial cells, suggesting that Müller glia are a source and sink for *Vegf* signals. Interestingly, *Mdk*, *Sdc2* and *Sdc4* are highly co-expressed in astrocytes and fibroblasts with *Sdc2* also expressed by pericytes, whereas *Sdc1* shows very little expression in these cells. The spatial profile of *Sdc1/2/4* shows preferential expression in spots overlapping with the outer retina (L4),

especially at 1PD and 3PD. *Mdk* spatial expression is also concentrated to the outer retina (L4) with intensity increased at 5PD. *Ptn* expression is highest in astrocytes and fibroblasts and more moderate in Müller glia, pericytes and vascular endothelial cells. Spatial distribution within the superior retina shows preferential expression of *Ptn* in the inner retina (L1, L2). The most prominently interacting receptor *Sdc3* is expressed in

**Fig. 5 | Gene expression changes originate in the inner retina and are mediated by key signalling networks. A** Representative sample illustrating the spots overlapping with the four retinal/choroid layers (left; Fig. 1Dii; Supplementary Fig. S8). Schematic showing the gene expression analysis paradigm: superior L1-L4: 1PD / 3PD / 5PD vs DR (middle). GO term enrichment analysis (Supplementary Data 5-Table S13, Supplementary Data 5-Table S14) of significantly differentially expressed genes ($p \leq 0.05$, FC ≥ 1.5; Supplementary Data S5-Table S12) in each layer in the superior retina. **B** CellChat[36] analysis showing strength of incoming (sink) and outgoing (source) signal in each layer at each time points for the top three signalling pathways. Cumulative signalling strength (top bar chart) and number of interactions for each ligand (right bar chart) are shown. The heatmap indicates relative signalling strength for each ligand. Signalling strength for all identified pathways are shown in Supplementary Fig. S11. **C** The inferred signalling network for VEGF, PTN and MDK ligands across the layers at each time point. Line colours indicate layer of origin and line thickness indicates relative strength of communication. **D** CellChat analysis showing significant receptor-ligand communication probabilities (heatmap) for VEGF, PTN and MDK interactions. Pairings with strong changes between time points are indicated in bold.

pericytes and vascular endothelial cells with spatial expression throughout the superior retina, although expression seems slightly reduced in the inner retina (L1). Contrasting, *Ncl* is highly expressed by all retinal cell types and, unsurprisingly, uniformly spatially distributed across the superior retina (Supplementary Fig. S12C&D).

## Discussion

To date, there have been several studies that have reported changes in the retinal transcriptome at a bulk[3,38–44] or single cell[45–54] level in various retinal degeneration mouse models, including PD. However, as retinal degenerations often exhibit regional pathology, these approaches have shown limited success in unveiling localised transcriptional changes, severely hampering our understanding of disease manifestation and progression, and the identification of potential therapeutic targets. Here we utilised Visium Spatial Gene Expression to gain novel insights into the location of molecular changes in the normal mouse retina and in response to photoreceptor degeneration. We characterise a number of previously unreported regional gene expression changes (inferior vs superior; within the central superior retina; across retinal layers) including a strong enrichment of upregulated genes within angiogenic and vascular pathways and identify angiogenic signalling networks not previously associated with this PD model. Further, we have developed the computational pipelines required for future investigations and created an online searchable data repository to enable easy, user-guided investigations of the spatial and temporal retinal gene expression changes associated with retinal degenerations: Spatial Transcriptomics of the Retina in Response to Photo-oxidative Damage (STRiPD), https://www.clearvisionresearch.com/stripd.

The pigmented mouse retina exhibits distinct regional variations in its structure, such as variances in S-opsin and L-opsin cone density[33,55,56] and differences in gene expression patterns both in normal physiological conditions and during disease states[28,55]. Despite considerable research efforts, the full extent and intricacies of these differences are still being elucidated. Overall, we found that the superior retina of the C57BL/6 J wildtype mouse naturally has some level of immune activation when compared to the inferior retina. Inflammation[2,4,19,20,57] is a major contributor to progressive retinal degenerations and our data confirms this by showing that inflammation was associated with degeneration, particularly at 5PD, with higher expression of known inflammatory genes such as *Igf1, Cd36, A2m, C3* and *Cfi* in the superior retina. Surprisingly, this dysregulation of the immune system was found to be accompanied by neovascular processes, which has not previously been reported (Fig. 2C, Supplementary Data S2-Table S4, Supplementary Data S2-Table S5A).

During healthy DR conditions, we detected a distinct gene expression profile in the superior retina. Firstly, we identified several genes upregulated in the superior homeostatic retina which are known to produce spatial protein gradients to facilitate dorsoventral patterning, axon routing (*Vax2, Efnb1* and *Efnb2, Nr2f2*;[58–60]) and dorsal choroidal vascular development (*Aldh1a1*;[61]) during retinal development. We suggest this reflects ongoing neuronal maintenance and repair in the superior retina to support the homeostatic state. Secondly, GO terms associated with genes that were more highly expressed in the superior retina were involved in processes typical for the retinal immune response.

In the healthy retina, microglial cells predominantly undertake immune surveillance responsibilities, aiding in cellular maintenance while promptly responding to insults or injuries, with Müller cells and

the RPE providing additional support[4]. Consequently, upregulation of genes associated with innate immune response pathways in the superior retina was not unexpected, given previous findings indicate similar responses during injury[62–66]. What was unexpected, is to find this in DR conditions, indicating that there might be an inherent necessity for the superior retina to consistently address low levels of immune modulation. However, it was particularly surprising to observe multiple processes typically associated with immune cell infiltration during late-stage degenerations occurring in the homeostatic superior retina, including but not limited to 'antigen processing and presentation' (GO:0019882), 'immunoglobulin-mediated immune response' (GO:0016064), and 'leukocyte-mediated immunity' (GO:0002443). These observed gene expression differences likely play a crucial role in the susceptibility of the superior retina to degeneration by maintaining stressors specific to this retinal region; which might be external, such as light exposure or mechanical stress, or internal, such as oxidative stress or inflammation. Therefore, this unique transcriptomic profile of the superior retina governs its response to these stressors, potentially resulting in increased vulnerability to degeneration. Understanding the molecular mechanisms underlying these gene expression differences is essential for elucidating the factors that drive the susceptibility of the superior retina to degenerative processes and developing targeted interventions to mitigate such susceptibility to external and internal stressors.

Additionally, the expression profile of the homeostatic superior retina showed an association with MHC class II-antigens (GO:0019886), which are present on the surface of antigen-presenting cells, such as macrophages and dendritic cells[67], but not present on microglia, the predominant immune population of the homeostatic retina[68]. We suggest that this could reflect the activation of RPE cells[69] and/or recruitment of the immune population from the underlying, highly vascularised choroid, which may also present spatial specificity[63]. The choroid supplies metabolic and immune support to the retina and contains many MHC-expressing lymphocytes, such as tissue macrophages, dendritic cells and mast cells[70]. In concordance, we identified higher levels of *Cd74*, a receptor for the macrophage migration inhibitory factor (MIF), in the superior retina. The CD74/MIF pathway plays a significant role in the innate immune system, mediating inflammation in response to injury and retinal degeneration[71]. Elevated CD74 levels in response to inflammation have been detected in retinal dystrophies but were restricted to activated microglia located near vessels that undergo vasoregression[72–74]. However, CD74/MIF can also act as a potent pro-angiogenic factor promoting pathological angiogenesis in several systems, including the retina[32] and imbalances could give rise to pathological angiogenic changes. We therefore suggest that the superior retina has a more unique and spatially defined inflammatory profile than previously suspected, possibly making it more susceptible to photoreceptor cell death and subsequent increases in the inflammatory response than the inferior retina.

Further evidence for a unique transcriptomic profile of the superior retina is the consistently increased expression of the autotaxin (ATX) enzyme (encoded by *Enpp2*) and the LPA receptor (encoded by *Lpar1*) in the homeostatic superior retina. The ATX enzyme and LPA receptor are required to maintain the structural integrity of the RPE and blood-retinal barrier (BRB)[75]. The BRB consists of the choroid, Bruch's membrane and RPE (outer BRB) and tight junctions between endothelial cells lining the retinal vasculature (inner BRB), which is important in maintaining the

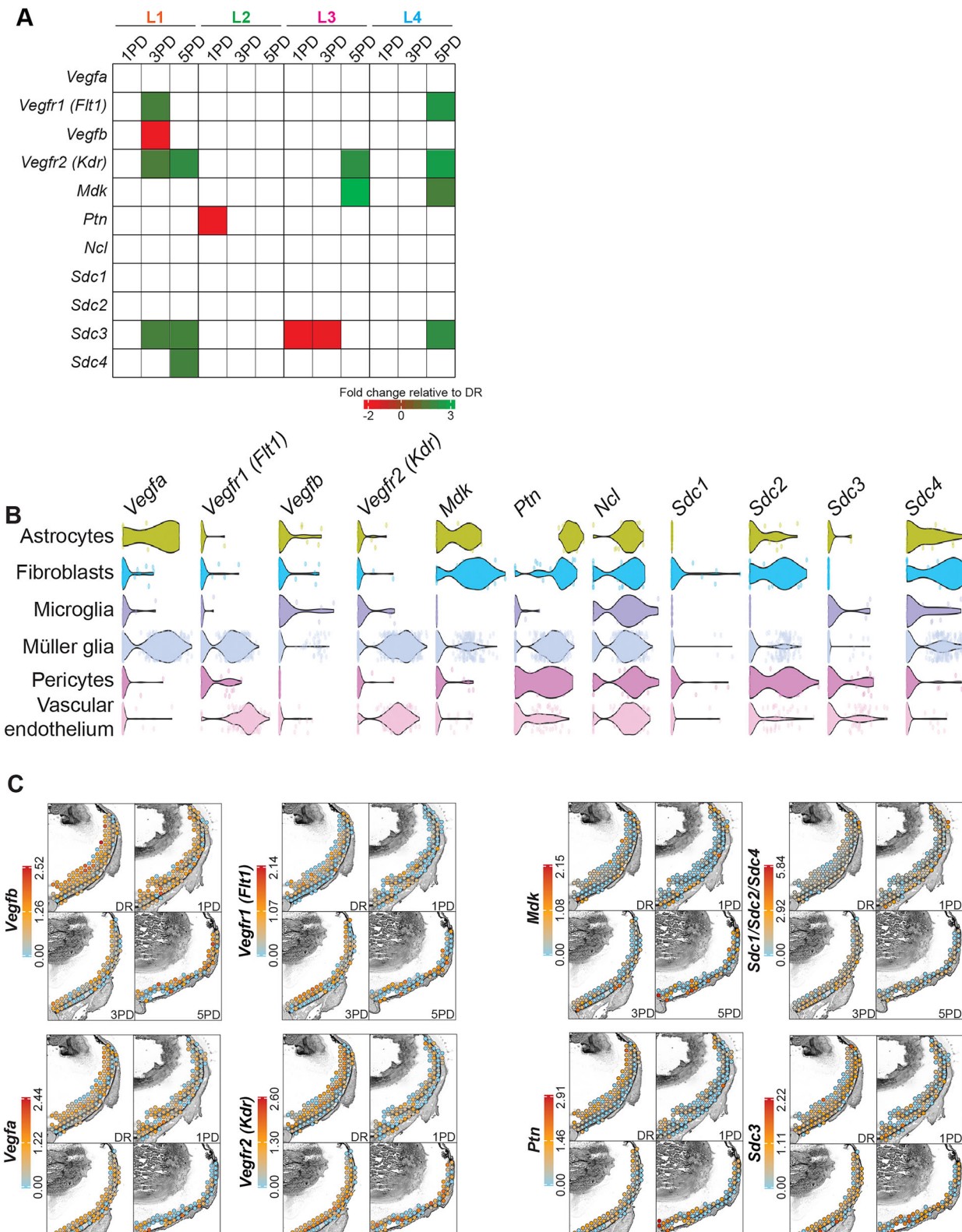

**Fig. 6 | Signalling network alterations are mediated by key retinal cell types.**
**A** Differential gene expression analysis (indicated by heatmap) in each retinal/
choroid layer at each time point relative to the DR control. Only significant ($p \le 0.05$)
changes are indicated. **B** Violin plots showing published single cell expression
profiles[37] of key signalling pathway genes across retinal cell types. Expression profiles
of all retinal cell types is shown in Supplementary Fig. S12C. **C** Per spot cumulative
gene expression level (indicated by heatmaps) of key ligand or receptor genes of the
VEGF, PTN and MDK signalling networks across the superior retina at all time
points. One representative replicate for each time point is shown.

immune privilege of the neural retina[76]. LPA acts as a potent growth factor that regulates retinal neurogenesis and RPE proliferation[77] but can also activate inflammatory and angiogenic pathways[78]. The ATX/LPA/LPAR network is normally tightly controlled and small changes to this balance could exacerbate the inflammatory and angiogenic effects of LPA[78,79].

In response to degeneration, the superior retina shows further distinct changes compared to the inferior retina (Fig. 2, Supplementary Data S2-Table S4, Supplementary Data S2-Table S5B-D). The top enriched GO pathways associated with genes upregulated in the superior at 1PD almost exclusively contained crystallin genes, which are small chaperones similar to heat-shock proteins. Crystallins exert numerous neuroprotective functions, especially in RPE cells, and are known to be upregulated in response to retinal pathologies[80–82]. Notably, loss of crystallin gene expression has been shown to reduce retinal ganglion cell survival during neurodegeneration[83]. Upregulating the expression of genes that preserve neuronal function is a rapid way to maintain neuronal health, especially if the need for neuronal support is transient, and could be a pathway for future therapeutic approaches.

At 3PD, and progressing into 5PD, we identified upregulation of genes associated with angiogenesis and the extracellular matrix. At 3PD this included genes whose dysregulation is known to alter cytokine signalling and angiogenesis, such as *Agt*, *Rras*, *Cav1*, *Ramp2*. Angiotensinogen (AGT) is part of the renin-angiotensin system (RAS) where angiotensinogen is converted by renin to angiotensin I (Ang I), which is then converted to Ang II. Generally, Ang II binds to the AT1 receptor resulting in vasoconstriction, whereas binding to AT2 has the opposite effect[84]. However, retinal Ang II function is multilayered, and several studies suggest that RAS hyper-activation in the retina induces inflammation, neurodegeneration, and neovascularisation in several retinopathies[85–87]. Inhibiting this pathway has shown some promise in reducing the negative impacts of hyperactive RAS[88,89]. Similarly, loss of *Cav1* has been shown to reduce pro-inflammatory cytokines, neovascularisation and BRB breakdown[90–92]. In contrast, loss of *Rras* has been directly associated with increased vessel permeability in neovascular disease[93], and loss of *Ramp2* led to enhanced neovascularisation[94]. This indicates a complex network of genes that could contribute to pathological neovascularisation. While there have been studies into neovascular effects in other rodent light damage models[21,63,95–98], the identification of neovascular processes has not been previously reported for the PD mouse model investigated here. However, both *Cav1* and *Ramp2* have been implicated in preventing fibrosis, with loss of *Ramp2* and increased expression of *Cav1* blocking epithelial-mesenchymal transition[94,99], suggesting that their contribution to neovascularisation is indirect. Indeed, previous findings have suggested that changes to RPE and choroid integrity might facilitate the growth of blood vessels into the retina, contributing to neovascularisation as seen in late-stage AMD[100,101]. This might suggest that, at 3PD, the superior retina initiates a transcriptional program to strengthen cellular structure and preserve integrity, which is not required in the inferior. The development of pathological neovascularisation might depend on the strength and success of this transcriptional remodelling. Taken together, this strongly suggests that the superior retina is inherently more susceptible to the effects of photo-oxidative damage compared to the inferior retina, likely because the insult/injury is spatially restricted. It was, however, surprising to uncover angiogenesis-related changes in the superior transcriptional profile compared to the inferior, as this indicates that this mouse model might recapitulate aspects of choroidal neovascularisation. Indeed, further research is needed to comprehensively unravel the underlying molecular mechanisms governing the differential susceptibility between the superior and inferior retina to degeneration. Investigating how spatial differences in gene expression, tissue morphology, and cellular microenvironment contribute to vulnerability to degeneration will be crucial.

Previous investigations into the photo-oxidative damage rodent models have demonstrated localised degeneration in the superior retina[2,21,63,102–104]. Our data support these previous studies by demonstrating an early oxidative stress response that is followed by increased inflammation at later timepoints (Fig. 3, Supplementary Data S3-Table S6, Supplementary Data S3-Table S7). However, we further show here a distinct angiogenic role for a specific region within the superior retina.

In retinal degenerations, prolonged periods of oxidative damage are known to be associated with upregulation of cytokine signalling and complement activation[66,105]. Broadly, genes upregulated in the superior retina in response to 1PD were most strongly associated with 'unfolded stress response' (GO:0006986), which included the transcription factors *Atf4*, a known effector of the integrated stress response (ISR), and *Atf3* and *Ddit3* (also known as *Chop*), both mediators of the endoplasmic reticulum (ER) stress response. ISR is a central coordinator of various stress signals including amino acid starvation, oxidative stress, and ER stress. These TF are known to elicit pro-apoptotic signalling and their inhibition has been shown to protect neurons and retinal ganglion cells from apoptosis[106–111]. Further, *Atf4* has been shown to induce *Mcp1* (also known as *Ccl2*) expression in microvascular endothelial cells, which promotes the infiltration of inflammatory cells[112] and increases VEGF levels in RPE cells[110]. At 3PD, we observed upregulation of genes associated with immune effector processes in the superior retina. Interestingly, we show upregulation of *Cd36*, which has been shown to be critical for balancing pro- and anti-inflammatory macrophages and reducing pathological blood vessel formation through interacting with TSP-1[113]. CD36 is also critically important for normal photoreceptor outer segment phagocytosis by RPE[114], with loss of CD36 leading to age-related photoreceptor and choroidal degeneration[115]. As expected, degeneration-associated processes in the superior retina appear to progress further at 5PD to establish immune cell activation and associated processes such as phagocytosis, cell migration and cytokine production.

Interestingly, many of these genes were preferentially upregulated in a region around 800 μm superior of the optic nerve head (Fig. 3C), which we termed R5. While previous work has shown targeted degeneration approximately around this area[2,21,63,102–104], this has not been explored transcriptionally. Thus, we employed computational methods to focus our analysis on R5 specifically, which revealed a complex angiogenic pattern across the time course (Fig. 4, Supplementary Data S4-Table S9E, Supplementary Data S4-Table S10).

Firstly, this showed that genes upregulated at 1PD are more strongly associated with GO terms involved in oxidative stress (GO:0034599, GO:1902882), and neuron death (GO1901215, GO:0070997). When delving further into the dysregulated transcripts, we found that genes with largely pro-angiogenic roles were downregulated in R5 at 1PD, including *Mef2c*, *Col1a2*, *Serpinf1* and *Pdgfrb*. *Mef2c* has been associated with neovascularisation in oxygen-induced retinopathy[116], whereas *Pdgfrb* has been associated with BRB breakdown in models of Alzheimer's disease and diabetic retinopathy[117,118]. Additionally, a circular RNA derived from *Col1a2* has been shown to be pro-angiogenic in diabetic retinopathy[119]. This suggests that combined downregulation of these genes elicits anti-angiogenic effects. Interestingly, *Serpinf1* (also known as *Pedf*) is a strong neurotropic factor and known inhibitor of retinal angiogenesis[120–122]. Thus, loss of *Serpinf1* expression may have an indirect pro-angiogenic effect by increasing inflammation leading to vascular changes[5]. Together, dysregulation of pro- and anti-angiogenic factors indicates a highly localised regulation of angiogenesis. This further suggests that at 1PD the retinal structure might already be compromised, which could be a source for the spread of degeneration across the superior retina. At 3PD, upregulated genes are associated with GO terms related to 'negative regulation of immune response' (GO:0050777) and 'regulation of angiogenesis' (GO:0045765). Similar to the whole superior retina, at 5PD upregulated genes are associated with cell migration (GO:0030335, GO:0050900), 'cytokine production' (GO:0001816) and many angiogenic GO terms, including vasculature development (GO:0001944, GO:1901342, GO:1904018) and regulation of angiogenesis (GO:0045765, GO:0045766). This suggests that in the PD model, R5 is a region susceptible to angiogenesis, possibly due to its susceptibility to neuronal cell death and BRB breakdown.

There are, of course, other factors that influence whether angiogenesis occurs, and many genes with angiogenic roles are found to be dysregulated, especially at 5PD, including *Adamts1, Cd36, Rras, Itgb1, Apoe, Lrp1, Pgf, Ang* and many others with established roles in pathological retinal angiogenesis[85,87,113,123]. This dysregulation is reflected in the GO term analysis that showed association with angiogenic pathways at both 3PD and 5PD. Across the time course, upregulation of a mix of pro- and anti-angiogenic genes occurs in R5, including *Tnfrsf1a, Fgf2* and *Klf2. Tnfrsf1a* knock-out mice have been shown to be highly susceptible to CNV lesion formation with increased immune cell infiltration[124], suggesting that *Tnfrsf1a* has an anti-angiogenic role. Additionally, *Klf2* is a known anti-angiogenic and anti-inflammatory factor in endothelial cells[125] and was exclusively upregulated in R5 across the time course. Interestingly, *Klf2* and *Mef2c*, which is downregulated across the time course, are transcription factors and downstream targets of the Erk5 pathway, which controls many aspects of endothelial cell functions, including inflammation and angiogenesis through VEGF signalling[125,126]. Accordingly, loss of *Mef2c* expression in endothelial cells reduced retinal neovascularisation[116]. Similar to published work[127], we show a significant downregulation of *Mef2c* in R5 at all time points, contrastingly however, *Erk5* and *Vegfa/b* show no significant dysregulation. Further, *Fgf2* has been shown to be essential for the development of pathogenic angiogenesis in CNV mouse models[128], although only when the tissue experienced extensive injury, suggesting that *Fgf2* levels are tightly controlled[129]. A recent study reported upregulation of pro-angiogenic pathways and associated neovascular morphological changes of the retina at seven days after recovery from acute light exposure[95]. This response was mediated through the upregulation of VEGF, VEGFR1/R2 and FGF, which we also show to be upregulated. Maybe not unsurprisingly, this suggests that the morphological development of neovascularisation is a time-dependent process. Given that our study examined the retina immediately following PD, our findings might have uncovered so far unknown retinal mediators of angiogenesis that could serve as early instigators of this later pathological neovascular process. Together, this data highlights that although pro-angiogenic genes are induced in a precise location, this does not necessarily lead to neovascular pathology. Future studies should investigate whether the observed changes in gene transcription translates to modulations in protein levels or localisation. However, as stated above, retinal angiogenesis has, to our knowledge, not been described as an immediate feature of the mouse PD model and therefore our data provide a platform for future work to identify key genes and potential therapeutic targets that suppress angiogenic gene expression, prior to the manifestation of vascular pathology.

As we identified R5 as a transcriptionally unique region, we suggest that identifying the cellular origin of these transcriptional changes is paramount for future studies into novel therapeutics. To achieve this within the confines of the resolution of the data, we analysed expression changes across individual layers (L1-L4) in the superior retina (Fig. 5). Amongst many of the changes identified, we found that genes downregulated in L4 at 3PD were significantly associated with 'extracellular matrix organisation' terms (Supplementary Data 5-Table S13). These include *Ccn2*, which modulates cell adhesion and motility and plays a role in blood vessel formation and regeneration[34], and several collagen type I, III and IV genes (*Col1a1, Col1a2, Col3a1, Col5a1* and *Col5a2*), which play an important structural role in Bruch's membrane and outer BRB integrity[35]. We suggest that this indicates pathological changes in the retinal structure initiated by the loss of RPE cells and subsequent breakdown of the outer BRB, resulting in infiltration of MHC-presenting cells from the choroid and enabling potential neovascular changes.

Using computational cell communication analyses tools, we show that the top retinal signalling networks include VEGF, midkine (MDK) and pleiotrophin (PTN). We found that, generally, all layers, except L3 (photoreceptors), serve as a either a signalling source or sink for at least one of these three networks, and show dramatic signalling probability changes for all three in response to the PD paradigm. However, there are additional subtle changes in each network for future exploration. For example, although generally a source for VEGF signalling, at 5PD L4 (RPE and

choroid) becomes a sink for signalling from L1 (ganglion cells) and L2 (interneurons). Similarly, although generally a source for MDK signalling, at 5PD L4 becomes a sink for signalling from all other layers. As we identified major cross-retinal signalling networks with key receptor-ligand interactions, we incorporated published single-cell sequencing data[37] to identify cell types that might participate in these networks (Fig. 6).

VEGF is an important factor in both physiologic and pathologic angiogenesis. Consequently, elevated VEGF is closely linked to several visual disorders such as diabetic retinopathy, exudative AMD and retinopathy of prematurity, and targeting this molecule is a commonly used therapy[130,131]. Further, a recent transcriptome-wide spatial-resolution study of neovascular human retinal and choroidal tissue[132] showed upregulation of angiogenesis-related genes and pathways, especially in endothelial and retinal pigment epithelial (RPE) cells, at the penumbra of degeneration. These gene expression changes were regulated by the upstream vascular endothelial growth factor (VEGF) and transforming growth factor *beta* (TGF-β1) signalling pathways offering new insights into already established gene expression profiles in the neovascular retina.

Considering the VEGF network, we show that L2 is the sink for VEGF signalling from all other retinal layers (including L2 itself), which increased in strength at 3PD (Fig. 5C). Conversely, expression of the VEGF receptor *Vegfr2* (*Kdr*) was upregulated in L1, L3 and L4 at 5PD, with expression of both *Vegfr1* (*Flt1*) and *Vegfr2* (*Kdr*) also upregulated in L1 at 3PD (Fig. 6A). However, VEGF is known to be extensively regulated post-transcriptionally to achieve delicate modulation of this molecule[133] and in our data, *Vegf* expression is generally not significantly modulated. This limits inferences of changes within this signalling network to the transcript levels of the VEGF receptors. Additionally, the NRP1 and NRP2 VEGFR co-receptors can also interact with the VEGF ligands to modulate a biological effect[121,134] and although we did not identify these as significantly contributing to the VEGF signalling network (Fig. 5D), *Nrp2* expression was upregulated in L2 and R5 at PD5 (Supplementary Data S4-Table S9, Supplementary Data S5-Table S12). Therefore, changes in receptor expression are likely not the sole mediator of retinal VEGF signalling. However, it is important to note that both VEGF receptors are highly expressed in Müller glia and vascular endothelial cells (Fig. 6B), which indicates a potential site of action. Müller glia traverse the entire retina with their cell bodies residing within the inner nuclear (interneuron) layer (L2), with endothelial cells residing within the retinal vasculature (L1 and L2), and the choroidal vasculature (L4). As our spatial data suggests that both VEGF receptors are preferentially expressed in L1 and L2 (Fig. 6C), we suggest that Müller glia and endothelial cells are the likely targets of VEGF signalling. It has been suggested that endothelial cells can initiate distinct transcriptional profiles in response to the same stimulus depending on their location[135]. In our context here, this might indicate that endothelial cells within the retinal vasculature respond differently to those in the choroidal vasculature, initiating distinct functional outcomes as recently reported for retinal angiogenic tip cells[136]. Interestingly, *Vegfa* is highly expressed in Müller glia but is also expressed in astrocytes in response to hypoxia[137]. Astrocytes reside within the interneuron layer (L2) but are also found in the inner retina (L1). As VEGF signalling originates from L2, Müller glia are most likely mediating this communication throughout the retina via *Vegfa*, similar to what was previously reported in the zebrafish retina[138], with a potential additional role for astrocytes.

Relative to VEGF, the heparin-binding growth factors MDK/PTN are poorly studied in retinal degenerations. MDK/PTN are neurotrophic factors, known to function in cell growth and migration, neuronal development as well as angiogenesis[139]. Here we show that L4 (RPE/choroid) is the source of MDK signalling across the retina (Fig. 5C), with *Mdk* strongly expressed in astrocytes and fibroblasts (Fig. 6B). We found that *Mdk* expression is upregulated in L3, L4 and R5 at 5PD (Supplementary Data S4-Table S9, Supplementary Data S5-Table S12), supporting our finding that L4 is the primary signalling source. Contrasting, L1 and L2 are the primary sources for PTN signalling across the retina, with *Ptn* strongly expressed in most retinal cell types (Supplementary Fig. S12A), but especially in

astrocytes and fibroblasts. Additionally, our spatial mapping suggests a preferential localisation of *Ptn* expression to L1 and L2 (Fig. 6C), supporting our finding that these layers are a source of PTN signalling. Although signalling strength of both MDK and PTN seems to increase at 1PD and 3PD, the expression of both transcripts is not altered at these time points, suggesting that changes in signalling strength might be mediated through their receptors.

As MDK and PTN ligands are ~50% homologous[140] their receptors, namely NCL and the SDC family, overlap making it more difficult to detangle these networks. The NCL receptor is ubiquitously expressed throughout the retina and across most retinal cell types (Supplementary Fig. S12), which might indicate that NCL plays a role in signal distribution across the retina. Indeed, we show that interaction of MDK and PTN with NCL evokes the most profound changes in communication probability at 1PD (Fig. 5D). Both MDK and PTN also interact with SDC1/2/4 (Fig. 5D), other potential mediators of the signalling network. Whereas *Sdc1/2* receptors showed no expression changes, *Sdc4* was upregulated in L1 at 3PD and R5 and 5PD (Supplementary Data S4-Table S9, Supplementary Data S5-Table S12), suggesting that this receptor could mediate the PTN signalling increase, at least from L1. Interestingly, we found that *Sdc3* interacts only with the PTN not the MDK receptor (Fig. 5D). Further, *Sdc3* shows a complex expression pattern (Supplementary Data S5-Table S12) with upregulation in L1 (at 3PD and 5PD) and L4 (at 5PD), but downregulation in L3 (at 1PD and 3PD). We also found that *Sdc3* is relatively lowly expressed in most retinal cell types (Supplementary Fig. S12A). Inhibition of PTN has been shown to reduce neovascularisation[141] and MDK was identified as an important modulator of retinal glial response to damage[142]. However, with this complex ligand expression pattern and hampered by the limited resolution, it is difficult to resolve how changes in PTN and MDK signalling networks establish across the retina, identify key cell types involved, and delineate functional outcomes. Some of these changes might be the key to the subtle signalling alterations we observed and with the addition of PSAP signalling, which also showed only subtle changes across the time course, these networks critically require further detailed investigation. Although anti-VEGF therapeutics have been successful, a large number of patients do not respond to these therapies. Thus, developing a better understanding of the MDK, PTN and PSAP angiogenic signalling networks in retinal degenerative diseases might lead to the development of novel therapeutic approaches.

In conclusion, this study comprehensively examined the transcriptional changes within the pigmented murine eye in response to PD, uncovering novel transcriptional aspects and ultimately providing a rich resource for future studies. In this study, we narrowed our focus to the retina to characterise the spatial transcriptome and its changes in response to photo-oxidative damage, which can be explored at STRiPD (https://www.clearvisionresearch.com/stripd). Plotting spatial gene expression brought into focus the locally distinct nature of gene expression changes in response to the PD time course, highlighting the need to incorporate spatial techniques with existing bulk or single-cell data. We showed inherent transcriptome differences in the superior retina, due to its unique structural requirements, which indicate a heightened immune regulation and extensive changes in response to degeneration. Surprisingly, we identified significant changes in angiogenesis-related pathways already at early time points, which has not been observed previously. Further, we highlight the potential for distinct cross-retinal communication of cytokines with known roles in angiogenesis. Finally, we and others have shown that, while the resolution of this technology does not yet compete with that of low-throughput spatial techniques that achieve single cell or sub-cellular resolution, the integration of spatial information into molecular analyses enables highly insightful discoveries. Consequently, our study has laid the groundwork and developed the computational tools required for future in-depth, high-resolution approaches and identified several molecular pathways requiring future exploration.

Although this data validates several well-established findings, including structural differences and the inflammatory response to photo-oxidative stress, this study has some limitations:

1) Targeted gene-specific validations of our cell layer- and retinal region-specific DE findings would be advantageous but were outside the scope of this work. Validations could include in situ hybridisation (ISH) techniques such as RNAscope, which offer increased resolution, to gain further insights and clarity into the specific cell types that drive the distinctly localised expression changes observed here.

2) Although we present strong computational evidence for additional angiogenic pathways at play in the retina, independent examination of these findings is required. Previous work has reported neovascular changes several days after the recovery from light stress, and including this aspect in a future analysis of the PD model would provide further insights. Additionally, the CNV mouse model[143], not yet fully established in our lab, would provide an ideal foundation to further investigate the contribution to neovascularisation of the angiogenic pathways identified here. Targeted depletion of genes such as *Mdk*, *Ncl* and the *Sdc* gene family members by siRNAs would bring to light their role in neovascularisation and suitability as a therapeutic target.

## Methods

### Photo-oxidative damage paradigm and tissue preparation

To induce photo-oxidative damage (PD), 8-week-old male litter-mate C57BL/6 J wilt-type mice were subjected to continuous 100 K Lux white LED light for 1, 3 and 5 days as previously described[2]. Dim-reared (DR) mice were used as control. Mice were allocated randomly to each experimental group and confounders were not controlled for. Experiments were performed twice for each time point ($n = 2$ animals per time point) to obtain two independent biological replicate tissue samples ($n = 2$) for each time point. The Visium spatial technology only facilitated the use of a single tissue section per biological sample. To obtain reproducible data, the experiments were conducted in biological duplicates. No exclusion criteria were set, and no animals or experimental replicates were excluded from the analysis.

Mice were cervically dislocated, and the superior region of both eyes marked. Eyes were then carefully enucleated, briefly washed in PBS and excess liquid removed using kimwipes. Eyes were placed in optimal temperature cutting medium (Tissue-Tek, Sakura) and then fresh frozen in an isopentane/liquid $N_2$ cooling bath. Embedded tissues were kept at −80 °C until further use. To ensure sufficient RNA integrity, 50 tissue sections (10 µm) from each time point were collected, RNA isolated using Trizol (Ambion) according to manufacturer's instructions and analysed using an Bioanalyser total RNA NanoChip (Agilent). The RIN for all samples was consistently above 7.0.

### Immunohistochemistry and TUNEL labelling

Following the PD paradigm, eyes were enucleated, fixed in 4% PFA for at least three hours and then washed over night in 10% sucrose in PBS. Eye were then embedded in optimal temperature cutting (OCT) medium and 10µm parasagittal sections (including the optic nerve head) analysed for immune cell infiltration, glial cell activation and cell death as previously described[63,102]. Microglia/macrophages were immunolabelled with primary mouse anti-chicken antibodies recognising ionised calcium-binding adaptor molecule 1 [IBA-1 (1:500, Aves Labs, IBA1-0020)] and secondary goat anti-chicken Alexa Fluor 488 (1:500, ThermoFisher Scientific, A11039). Glial cell activation was analysed by immunolabelling with primary antibodies recognising glial fibrillary acidic protein [GFAP (1:500, Invitrogen, MA5-12023)] and secondary goat anti-mouse Alexa Fluor 568 (1:500, ThermoFisher Scientific, A11031). Cell death was determined by terminal deoxynucleotidyl transferase (dUTP) nick-end labelling (TUNEL) assay using the Click-iT Plus TUNEL Kit (ThermoFisher Scientific, C10618) following manufacturers instructions. Sections were counterstained with Hoechst and imaged using a Nikon confocal microscope at 20x magnification.

## Spatial Transcriptomics workflow

For all experiments, embedded tissue was sectioned at the parasagittal plane at 10μm thickness. We utilised the Visium Tissue Optimisation and Spatial Gene Expression Kits (10X-1000200; 10X Genomics) for this work, following the most up-to-date manufacturer's demonstrated protocols.

*Tissue Permeabilization Optimisation* was performed using dim-reared (DR) and 5-day PD (5PD) tissue with HeLa cell total RNA extract as positive control. Serial sections were placed on the TO slides, tissue permeabilization optimisation performed and the tissue imaged using an Axio Scan.Z1 slide scanner (Zeiss), imaging all eight capture areas with the same settings (Supplementary Supplementary Fig. S1C). As recommended, we analysed the strength of the fluorescent signal combined with clarity and contrast across the section, especially in the retinal region as our ROI, and determined that 12 and 18 minutes produced good results, with no obvious difference between the two tissues. We reasoned that the integrity of the eye tissue would be largely unaffected by the PD paradigm, except a small region in the superior retina, and thus tissue from 1PD and 3PD would likely behave similarly to the DR and 5PD tissue. As 18 minutes produced slightly more contrast in the retinal region, especially the RPE, we chose this tissue permeabilization time for subsequent experiments (Supplementary Fig. S1C).

*Spatial Gene Expression* was performed using right eye sections from all four time points using biological duplicates from two independent experiments. One section from each time point was place within the capture area of the ST slide, stained with eosin and hematoxylin (H&E) followed by imaging (Supplementary Fig. S1D), tissue permeabilization (18 minutes) and Illumina sequencing library preparation. Tissue placement for the second replicate was incomplete and the slides were subjected to tissue removal according to manufacturer protocols before repeating tissue placement and completing the workflow. Sequencing libraries were prepared according to the protocol using a different index for each sample from the dual index kit TT set A (10X-1000215; 10X Genomics) to allow multiplexing. ST libraries were multiplexed, pooled and sequenced using the NovaSeq 6000 SP100 flow cell (Illumina), acquiring 2 x 50 bp reads. Read-producing spots were overlapped with the tissue images using the Space-Ranger software (10X Genomics).

## Computational analyses

**Read processing, quality control and spot clustering.** A total of 5880 spots were captured across all eight samples and the gene and spot count matrices obtained from the SpaceRanger software (10X Genomics) following alignment to the mouse genome (mmu-10). The gene and spot count matrices for all eight samples were then pre-processed to remove low-quality spots. Firstly, we assess RNA diffusion rates utilising the SpotClean package[23], which showed that on average 16% of UMI's originate from a neighbouring spot (Supplementary Fig. S2A). The largest rate of contamination occurred in spots that are either not covered by tissue or overlap the lens and vitreous chamber (Supplementary Fig. S2B). Secondly, we defined low-quality spots by the number of features (nFeature < 500) and/or mitochondrial expression levels (percent.mito > 30%), removing a total of 1803 spots from the analysis. Incidentally, spots identified by either approach largely overlapped and thus we only included low-quality filtering in our pipeline. We then analysed the reproducibility between the biological replicates, which showed exceptional correlation ($r \geq 0.98$; Supplementary Fig. S2C) with no significant differential expression between the replicates (Supplementary Data S1-Table S1, Supplementary Fig. S2D).

Initially we used the STUtility package[144] to integrate expression with H&E images (Supplementary Fig. S3 & S4) obtaining 20 clusters. However, we noticed that not all clusters were present in all samples. Thus, we explored stLearn[25], which produced superior results. We performed normalisation and integration with morphology using stLearn[25]. High-level features were extracted from each H&E image using the stlearn.pp.extract_feature function and expression profiles within each sample were normalised using the st.spatial.SME.SME_normalize function. The

eight normalised count matrices were then concatenated and scaled based on the topmost variable features. We utilised the Harmony package[26] to remove batch effects before performing principal components analysis (PCA) and UMAP dimension reduction[145]. Spot were clustered by expression using the Leiden algorithm[146] and the results visualised by PCA of the reduced data using the sc.pl.umap function. The clustered spots were then mapped back to their position of origin.

**Marker gene expression and spot deconvolution.** To identify cluster-specific marker gene expression, we used the FindMarkers function from the Seurat v3 package[24] followed by performing Welch *t*-tests on SME normalised counts for genes between all pairwise clusters. The top 5 genes specific for each cluster with the lowest *p*-value were deemed marker genes and their expression visualised in a heatmap. To determine the cell type contributing to the expression profile of each spot within the four retinal clusters, we integrated publicly available single cell sequencing data[27] using the scatterPie function from the Spotlight package[147]. As we focussed on retinal cell types, we removed vascular and endothelial cells before integration. The raw read counts from the single cell data were log-transformed and the 3000 most variable genes extracted using the getTopHVGs function. Additionally, we used the scoreMarkers function to extract marker genes based on the published cell type identification[27]. Genes describing cell identities were kept and each cluster down-sampled by randomly selecting 100 cells to represent each cell-type. The processed data was then used to train a non-negative matrix factorization (NMF) regression model using the runNMF function. Subsequently, each spot was deconvoluted using a non-negative least square (NNLS) method based on the NMF model and then visualised using the plotTopicProfiles function.

**Defining retinal clusters and regions.** Following stLearn spot clustering, four clusters were identified that reflected the distinct retinal layers by gene expression profiles and tissue localisation. To analyse layer-specific characteristics, these clusters were further refined by removing spots located clearly outside the retinal tissue (see Supplementary Fig. S8).

The impact of the photo-oxidative damage model is greatest in the central superior region of the retina (Supplementary Fig. S1A&B). To reflect this in our analysis paradigm, we used an innovative retinal segmentation approach. The expression of known inflammatory genes was plotted for each spot overlapping the retina across all eight samples. Utilising the Zen software (Zeiss), we then measured the distance from optic nerve head to the centre of the superior lesion site in the 5PD time point and identified the equivalent region in the other samples. The superior retina was then divided into three equidistant lengths, which was then mirrored for the inferior retina.

**Differential gene expression analyses.** Differential gene expression (DE) analysis between clusters was performed using the pseudo-bulk method within the edgeR package[148]. For each cluster of each biological replicate, the summed raw gene counts were used as the bulk sample for comparison, therefore multiple spots contribute to the gene expression count for each cluster in each replicate sample. Leveraging the pseudo-bulk approach allowed us to utilise traditional analysis tools developed for bulk RNA-seq data, which are well-established and statistically powerful. Further, this accounted for biological variability, as we have duplicate data ($n = 2$), which improved the reliability of our differential expression analysis results compared to methods like the Wilcoxon test, commonly used in studies without replicates. Then, non-specific filtering was performed to remove clusters with spot numbers less than 10. After filtering, the raw gene counts were normalised using the calcNormFactors function based on the library size and then scaled with edgeR::scale.

Within the retina, DE analysis was carried out using the edgeR package[148]. Several contrast matrices defining the spot groups for comparison were constructed. Using these matrices, the gene counts were fitted into a quasi-likelihood negative binomial generalised log-linear dispersion

model using the glmQLFTest function. We applied the Benjamini-Hochberg procedure to control the false discovery rate (FDR) for multiple hypothesis testing. To identify DE genes which are most likely to play critical roles, we utilised $p$-value < 0.01 and fold change > 1.5 criteria for down-stream analyses. We extracted the $p$-values using the topTags function and DE genes (Log$_2$FC ≥ 0.585, $p$ ≤ 0.05) were visualised using volcano plots.

**Pathway enrichment analyses and reduction.** Significantly differentially expressed genes were analysed for Gene Ontology (GO) over-representation using the enrichGO function from the clusterProfiler package[149]. The top 20 significantly enriched pathways (FDR ≤ 0.05) were subjected to hierarchical clustering using the pairwise_termsim function and visualised as dotplots using the treeplot function. Reduction and visualisation of GO terms (REVIGO[29]) was performed to cluster related terms and identify major pathways associated with differentially expressed genes. For visualisation, GO terms were clustered based on their log2 transformed $p$-value enrichment score at each timepoint studied.

**Plotting within spot gene expression.** For each spot, the SME normalised counts for each gene contained within the enriched pathway was summed and then plotted as a heatmap across all eight samples. One replicate sample is shown for each pathway.

**CellChat analysis.** Cell-cell communication dynamics were inferred using the CellChat package[36]. The CellChatDB.mouse database was used as the reference for the identification of overexpressed ligands and receptors across the retinal/choroid layers. Identified ligands/receptors were then used to deduce the layer-specific communication probabilities and infer the resulting cross-layer communication networks.

**Reporting summary**
Further information on research design is available in the Nature Portfolio Reporting Summary linked to this article.

## Data availability
All source data generated during this study are available at the NCBI Gene Expression Omnibus database under accession number GSE248517. All regional analyses, including spatial gene expression plotting of candidate genes can be interactively explored at (https://www.clearvisionresearch.com/stripd) STRiPD[150]. All other data are available from the corresponding authors on request. The source code developed in this project is available on Github (https://github.com/jiayuwen/Spatial_Retina). The analysed data can be explored and visualised (http://www.clearvisionresearch.com/STRiPD)[150], including but not limited to all comparisons described here. The website facilitates spatial gene expression plotting of a gene or gene group of interest, which can be downloaded.

## Code availability
All custom code developed during this study is available at https://github.com/jiayuwen/Spatial_Retina.

## Abbreviations:

| | |
|---|---|
| PD | photo-oxidative damage |
| 1PD | 1 day of photo-oxidative damage paradigm |
| 3PD | 3 days of photo-oxidative damage paradigm |
| 5PD | 5 days of photo-oxidative damage paradigm |
| DR | dim-reared control |
| AMD | age-related macular degeneration |
| RPE | retinal pigment epithelium |
| VEGF | vascular endothelial growth factor |
| MDK | midkine |
| PTN | pleiotrophin |
| PSAP | prosaposin |
| L1 | retinal spot layer 1; ganglion cell layer |
| L2 | retinal spot layer 2; inner nuclear layer |
| L3 | retinal spot layer 3; photoreceptors |
| L4 | retinal spot layer 4; RPE/choroid |

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

## Acknowledgements

The authors acknowledge the instruments and expertise of Microscopy Australia at the Centre for Advanced Microscopy, Australian National University, a facility enabled by NCRIS and university support. The authors would like to thank the ACRF Biomolecular Resource Facility (JCSMR, ANU) and the ANU Bioinformatics Consultancy for sequencing and support in integrating spatial transcriptomics sequencing data with the SpaceRanger software. This study was supported by the National Health and Medical Research Council of Australia (2020/GNT2002239, U.S., R.N.), the Australian Research Council Center of Excellence for the Mathematical Analysis of Cellular Systems (J.W.; CE230100001, L.L.), The Gretel and Gordon Bootes Medical Research and Education Foundation (U.S., R. A.-B., A.S.), the ANU Translational Fellowship (R.N.) and the NSW Health RNA Future Leader Program (U.S.). The funding bodies had no role in study design, data collection, analysis, or interpretation.

## Author contributions

U.S. and R.N. conceptualised and designed the study. U.S. and L.L. performed molecular experiments and collected the data. A.S. performed imaging. L.L. performed bioinformatics analyses. A.V.C. supported bioinformatics analyses. U.S., A.V.C. and R.N. interpreted the data. R.A.B. prepared the graphical abstract. U.S. prepared the figures and wrote the manuscript. R.A.B., K.V., M.M. and R.N. provided comments and edited the manuscript. J.W. and R.N. provided supervision. U.S., A.S., L.L., J.W. and R.N. acquired funding. All authors read and approved the final manuscript.

## Competing interests

The authors declare the following competing interest: RN is a member of the scientific advisory board of EyeCo. No other competing interests exist.

## Ethics approval and consent to participate

All animal experiments were conducted in accordance with the ARVO Statement for Use of Animals in Ophthalmic and Vision Research and with approval from the ANU Animal Experimentation Ethics Committee (Ethics ID: A2020/41). We have complied with all relevant ethical regulations for animal use. 8-week-old male litter-mate C57BL6/J wild-type mice were used for all experiments.
