## [Transparent Peer Review file · Communications Biology]

Spatial transcriptomics reveals regionally altered gene expression that drives retinal degeneration.

Corresponding Author: Dr Riccardo Natoli

This manuscript has been previously reviewed at another journal. This document only contains information relating to versions considered at Communications Biology.

Version 0:

Reviewer comments:

Reviewer #1

(Remarks to the Author)

Schumann and colleagues performed a spatial transcriptomic analysis on the eyeball in a retinal degeneration mouse model. Their study focused on analyzing retina region and identified the molecules involved in retinal degeneration. The methodology described in the manuscript appears technically sound, and analyses conducted were comprehensive and thorough. However, there are a few points that should be addressed as outlined below:

- 1) In figure 2, authors compared the superior and inferior retina to identify genes highly expressed in superior retina and indicated a higher level of inflammation in superior retina. If there were significant structural or cellular differences between superior or inferior retina, it would be expected to identify a considerable number of differentially expressed genes across all samples. However, as shown in fig 2a, there are limited common genes, and the enriched GO at each time points are all distinct (fig 2c). Additionally, they didn't account for the multiple testing issue in differential expression analysis (fig 2b). If a procedure of multiple testing correction had been applied, the number of identified differentially expressed genes would most likely have been substantially reduced, and the current conclusions might be over-interpreted.
- 2) Authors employed hierarchical clustering to display the enriched GO terms (such as Fig 2c and Fig 3c). In the method section, it is described that the hierarchical clustering was conducted based on the semantic similarity among GO terms. Nevertheless, it is perplexing why terms with similar semantics, such as "vasculogenesis", "angiogenesis", and "regulation of vasculogenesis" in Fig 2c, were arranged into distinct clusters.
- 3) The authors claimed to have identified a novel angiogenic pathway in retinal degeneration. However, further validation is lacking. It is advisable to conduct additional experiments, such as utilizing RNAscope or IHC staining, to validate the expression of key genes implicated in angiogenesis and to elucidate which specific cell type initiates the angiogenic process.

Reviewer #2

(Remarks to the Author)

Summary

Schumann et al. used the visium spatial gene expression technology to understand the spatiotemporal changes in retinal gene expression during progressive retinal degeneration in mice. This study revealed that the immunity related genes are highly enriched in the superior retina of both healthy and degenerating mouse retina. The study also showed highly localized molecular changes in injured retina, which may serve as novel therapeutic targets for treating ocular diseases. The manuscript is well written and provided additional information on regional transcriptome changes during retinal degeneration.

My specific comments are listed below:

1. For the benefit of researchers not working in field, please include a figure illustrating the histopathological changes of the retina across the PD time course. Please use published data or in house data to highlight, changes in inflammatory response, cell death, vasculature, etc. during the progression of PD.
2. It would be great to confirm the changes in key gene expression using other methods, such as RNA scope, at different stages of the PD progression.

3. Further characterization on new targets, such as MDK, PTN and PSAP, should be performed, at least to demonstrate their association with angiogenesis.

4. Discussion session: please highlight weaknesses of the study.

Version 1:

Reviewer comments:

Reviewer #1

(Remarks to the Author)

All comments have been addressed, and I have no further comments.

Reviewer 1:

Summary

Schumann et al. used the visium spatial gene expression technology to understand the spatiotemporal changes in retinal gene expression during progressive retinal degeneration in mice. This study revealed that the immunity related genes are highly enriched in the superior retina of both healthy and degenerating mouse retina. The study also showed highly localized molecular changes in injured retina, which may serve as novel therapeutic targets for treating ocular diseases.

The manuscript is well written and provided additional information on regional transcriptome changes during retinal degeneration.

We thank the reviewer for the positive response to our study. Their specific concerns are addressed below.

My specific comments are listed below:

1. For the benefit of researchers not working in field, please include a figure illustrating the histopathological changes of the retina across the PD time course. Please using published data or in house data to highlight, changes in inflammatory response, cell death, vasculature, etc. during the progression of PD.

We have included an additional Supplementary Figure (new S1A) to outline the progressive changes in immune cell infiltration (IBA positive cells), which illustrates inflammation, cell death (TUNEL ositive cells) alongside the retinal morphology (superior photoreceptor (ONL) layer thinning already demonstrated (new S1B) across the PD time course (L:98-102).

We further have included additional references to published work that illustrates these morphological changes across different PD paradigms (Bian et al 2016, Garcia-Ayuso et al 2011 and Marc et al 2008; L:98-102).

2. It would be great to confirm the changes in key gene expression using other methods, such as RNA scope, at different stages of the PD progression.

We thank the reviewer for this suggestion and agree that this would be a useful addition.

RNAscope or *in situ* localisation studies are extensive follow-up experiments and at this stage are outside the scope of this manuscript. We have included a paragraph on the conclusion section outlining that validation experiments should be part of a future study (L:819-823). We have also added a section outlining the weaknesses of the study (L:817-832), such as lack of secondary validation, which is outside the scope of this work.

Although this technique is novel, our extensive development of computational analytical pipelines along with biological duplicate data, has robustly identified known biological pathways associated with retinal degenerations (inflammation and complement activation for instance), which we believe validates the dataset on its own. In depth analysis and validation of individual molecules will require extensive further work outside of the scope of this study.

3. Further characterization on new targets, such as MDK, PTN and PSAP, should be performed, at least to demonstrate their association with angiogenesis.

We thank the reviewer for this suggestion. We agree that these pathways are a novel discovery in the PD model, and we explicitly state in the current manuscript that these genes and pathways should be characterised further (L:787-790). Angiogenesis is not normally associated with the PD model and thus it was surprising to uncover these pathways in our study, as we stress in the manuscript (L:587-590; L:676-679). However, a recent study by Tisi et al (2020; 10.1038/s41598-020-63449-y) showed retinal neovascular changes after 7 days of recovery from an acute light stress. This might suggest that morphological changes require time and possibly other factors only present when the stress is removed. Given that our study examines the retina immediately following the damage paradigm, our findings contribute to this body of literature by identifying novel retinal mediators of angiogenesis that may serve as early instigators of the angiogenic process, possibly uncovering the pathways that eventually lead to pathological retinal neovascular growth. We suggest that these angiogenic pathways might play a thus far unknown role that could provide further insight into how angiogenesis develops in retinal degenerative diseases. We have added a paragraph in the discussion of the manuscript to clarify this (L:672-780).

However, the CNV mouse model (doi: 10.1016/S0002-9440(10)65753-7) would be more suited to the direct investigation of these pathways and their role in angiogenesis in the retina and as such these experiments are extensive, outside the scope of this work and should be part of a follow up study. We have included a paragraph in the manuscript to outline this (L:824-832).

4. Discussion session: please highlight weaknesses of the study.

We have added a section that outlines the weaknesses of this work (L:817-832):

"Although we validate several well-established findings, including structural differences and the inflammatory response to photo-oxidative stress, this study has some limitations.

1) Targeted gene-specific validations of our novel cell layer- and retinal region-specific DE findings would be advantageous but were outside the scope of this work. Validations could include *in situ* hybridisation (ISH) techniques such as RNAscope, which offer increased resolution, to gain further insights and clarity into the specific cell types that drive the distinctly localised expression changes observed here.

2) Although we present strong computational evidence for additional angiogenic pathways at play in the retina, independent examination of these findings is required. Previous work has reported neovascular changes several days after the recovery from light stress, and including this aspect in a future analysis of the PD model would provide further insights. Additionally, the CNV mouse model (143), not yet fully established in our lab, would provide an ideal foundation to further investigate the contribution to neovascularisation of novel angiogenic pathways presented here. Targeted depletion of genes such as *Mdk*, *Ncl* and the *Sdc* gene family members, by siRNAs would bring to light their role in neovascularisation and suitability as a therapeutic target."

Reviewer 2:

Schumann and colleagues performed a spatial transcriptomic analysis on the eyeball in a retinal degeneration mouse model. Their study focused on analyzing retina region and identified the molecules involved in retinal degeneration. The methodology described in the manuscript appears technically sound, and analyses conducted were comprehensive and thorough.

We thank the reviewer for their positive comments on our study. We have addressed their specific concerns below.

However, there are a few points that should be addressed as outlined below:

1) In figure 2, authors compared the superior and inferior retina to identify genes highly expressed in superior retina and indicated a higher level of inflammation in superior retina. If there were significant structural or cellular differences between superior or inferior retina, it would be expected to identify a considerable number of differentially expressed genes across all samples. However, as shown in fig 2a, there are limited common genes, and the enriched GO at each time point are all distinct (fig 2c).

We thank the reviewer for this comment. We show in Figure 1F that, although differential gene expression between the superior and inferior retina is driven by genes that reflect their morphological differences, the number of DE genes seems moderate. In fact, we identify 145 DE genes between the superior and inferior retina across all samples (Table S4 column log2FC_ALL), which is a considerable number. When considering the differences between the two regions at DR conditions (Table S4 column log2FC_DR), we identify 253 DE genes which are predominantly associated with inflammation (Table S5A). Together, this suggests that DE genes between superior and inferior is a reflection of the different functions of these two distinct retinal regions.

The reviewer suggests that there is only a small overlap of DE genes across all time points, we identified 24 genes differentially expressed in at least 3 time points (Figure 2A), of which all but 2 are also differentially expressed in the superior retina overall (Table S4), suggesting that they do reflect structural and functional differences. Further, the reviewer suggested that there is little overlap between enriched GO terms in genes upregulated in the superior. This is not quite the case. Figure 2C only shows the top 20 enriched GO terms, as is stated in the legend, for each time point. The full list of enriched GO terms is shown in Table S5, which shows extensive overlap across all time points. In total 96 GO terms are enriched in upregulated genes at at least 2 time points with 7 GO terms enriched in at least 3 time points.

Additionally, they didn't account for the multiple testing issue in differential expression analysis (fig 2b). If a procedure of multiple testing correction had been applied, the number of identified differentially expressed genes would most likely have been substantially reduced, and the current conclusions might be over-interpreted.

We acknowledge the concern regarding multiple testing correction. Our study included two biological replicates, which is an improvement over many similar spatial transcriptomics studies that often do not include replicates due to cost constraints. The inclusion of replicates allowed us to perform a more robust differential expression analysis using a pseudo-bulk approach.

Pseudo-bulk analysis involves aggregating single-cell or spatial transcriptomics data across cells or spots within each replicate to create a bulk-like profile. This method enables the application of traditional bulk RNA-seq analysis tools, which are well-established and statistically powerful. By leveraging the pseudo-bulk approach, we accounted for biological variability and improved the reliability of our differential expression results compared to methods like the Wilcoxon test, commonly used in studies without replicates.

We applied the Benjamini-Hochberg procedure to control the false discovery rate (FDR) for multiple hypothesis testing. The application of FDR correction indeed reduced the number of significant differentially expressed genes, which was expected due to the inherent variability between the limited number of replicates. Despite the reduced number of significant genes, our main goal was to examine the relevant pathways associated with the top differentially expressed genes. Therefore, we used a criterion of p-value < 0.01 and fold change > 1.5 to identify these top-ranked genes. This approach allowed us to focus on genes with the most substantial changes in expression, which are more likely to play critical roles in the observed regional or layer's differences. We have added a paragraph in the methods sections that outlines this clearly (L:957-962; L:968-971).

2) Authors employed hierarchical clustering to display the enriched GO terms (such as Fig 2c and Fig 3c). In the method section, it is described that the hierarchical clustering was conducted based on the semantic similarity among GO terms. Nevertheless, it is perplexing why terms with similar semantics, such as "vasculogenesis", "angiogenesis", and "regulation of vasculogenesis" in Fig 2c, were arranged into distinct clusters.

Thank you for your insightful observation. The hierarchical clustering depicted in Figure 2C was performed based on the log p-values of enrichment at each time point, as indicated on the x-axis, rather than solely on the semantic similarity of the GO terms. This approach can result in terms such as "vasculogenesis," "angiogenesis," and "regulation of vasculogenesis" being placed into distinct clusters due to differences in their enrichment patterns over time. We have clarified this in the methods section (L:991-993).

3) The authors claimed to have identified a novel angiogenic pathway in retinal degeneration. However, further validation is lacking. It is advisable to conduct additional experiments, such as utilizing RNAscope or IHC staining, to validate the expression of key genes implicated in angiogenesis and to elucidate which specific cell type initiates the angiogenic process.

We thank the reviewer for this suggestion. We explicitly state in the manuscript that these pathways are a novel discovery in the PD model, and that further experiments should be conducted to gain further insights (L:779-782). However, a recent study by Tisi et al (2020; 10.1038/s41598-020-63449-y) showed retinal neovascular changes after 7 days of recovery from an acute light stress. This might suggest that morphological changes require time and possibly other factors only present when the stress is removed. Given that our study examines the retina immediately following the damage paradigm, our findings contribute to this body of literature by identifying novel retinal mediators of angiogenesis that may serve as early instigators of the angiogenic process, possibly uncovering the pathways that eventually lead to pathological retinal neovascular growth. We suggest that these angiogenic pathways might play a thus far unknown role that could provide further insight into how angiogenesis develops in retinal degenerative diseases. We have added a paragraph in the discussion of the manuscript to clarify this (L:672-780).

We agree that RNAscope or other *in situ* localisation studies would add valuable insights, however these should be part of a future study in depth study and we have included a paragraph to the conclusion section outlining that (L:819-823). We have also added a section outlining the weaknesses of the study (L:817-832), such as lack of secondary validation, which is outside the scope of this work.

Additionally, as stated in the manuscript, angiogenesis is not normally associated with the PD model (L:587-590; L:676-679). For stringent characterisation of these pathways, validation and follow-up experiments should be conducted using the CNV mouse model (doi: 10.1016/S0002-9440(10)65753-7) and we have added a section outlining this (L:816-821). These experiments are extensive, outside the scope of this work and should be part of a follow up study.